# Analysis of SMAD1/5 target genes in a sea anemone reveals ZSWIM4-6 as a novel BMP signaling modulator

Paul Knabl[1,2†], Alexandra Schauer[1†‡], Autumn P Pomreinke[3§], Bob Zimmermann[1], Katherine W Rogers[3#], Daniel Čapek[4], Patrick Müller[3,4], Grigory Genikhovich[1*]

[1]Department of Neurosciences and Developmental Biology, University of Vienna, Vienna, Austria; [2]Vienna Doctoral School of Ecology and Evolution (VDSEE), University of Vienna, Vienna, Austria; [3]Friedrich Miescher Laboratory of the Max Planck Society, Tübingen, Germany; [4]University of Konstanz, Konstanz, Germany

**\*For correspondence:**
grigory.genikhovich@univie.ac.at

[†]These authors contributed equally to this work

**Present address:** [‡]Institute of Science and Technology Austria, Klosterneuburg, Austria; [§]Department of Molecular Genetics, Institute of Biology, University of Hohenheim, Stuttgart, Germany; [#]Division of Developmental Biology, Eunice Kennedy Shriver National Institute of Child Health and Human Development, NIH, Bethesda, United States

**Competing interest:** The authors declare that no competing interests exist.

**Abstract** BMP signaling has a conserved function in patterning the dorsal-ventral body axis in Bilateria and the directive axis in anthozoan cnidarians. So far, cnidarian studies have focused on the role of different BMP signaling network components in regulating pSMAD1/5 gradient formation. Much less is known about the target genes downstream of BMP signaling. To address this, we generated a genome-wide list of direct pSMAD1/5 target genes in the anthozoan *Nematostella vectensis*, several of which were conserved in *Drosophila* and *Xenopus*. Our ChIP-seq analysis revealed that many of the regulatory molecules with documented bilaterally symmetric expression in *Nematostella* are directly controlled by BMP signaling. We identified several so far uncharacterized BMP-dependent transcription factors and signaling molecules, whose bilaterally symmetric expression may be indicative of their involvement in secondary axis patterning. One of these molecules is *zswim4-6*, which encodes a novel nuclear protein that can modulate the pSMAD1/5 gradient and potentially promote BMP-dependent gene repression.

## Editor's evaluation

This important work presents a systematic survey of downstream target genes of the BMP pathway during body-axis establishment of the cnidarian *Nematostella vectensis*. Combining genomic approaches and genetic manipulations across species, the authors present convincing evidence that Zswim4-6 acts as a negative feedback regulator of BMP activity whose function is evolutionarily conserved. Thus, this work will be of interest to researchers in both developmental biology and evo-devo.

## Introduction

The clade Bilateria unites animals with bilaterally symmetric body plans that are determined by two orthogonally oriented body axes, termed the anterior-posterior (A-P) axis and the dorsal-ventral (D-V) axis. These body axes form a Cartesian coordinate system in which the location of different morphological structures is specified by gradients of morphogen signaling (*Niehrs, 2010*). Notably, bilaterality is also observed in representatives of a single animal clade outside of Bilateria - their evolutionary sister group Cnidaria (*Berking, 2007*; *Finnerty et al., 2004*). Common to all cnidarians is the formation of an oral-aboral (O-A) body axis which is patterned by Wnt/β-catenin signaling (*Kraus et al., 2016*; *Lebedeva et al., 2021*; *Lee et al., 2007*; *Marlow et al., 2013*; *Momose et al., 2008*; *Momose and Houliston, 2007*; *Niedermoser et al., 2022*; *Wikramanayake et al., 2003*).

However, in contrast to Medusozoa (jellyfish and hydroids), Anthozoa (sea anemones and corals) have an additional secondary, 'directive' body axis (*Figure 1A*), which is patterned by bone morphogenetic proteins (BMPs, *Figure 1B*) signaling (*Finnerty et al., 2004*; *Saina et al., 2009*). While BMP signaling was also shown to regulate patterning of the D-V axis in Bilateria (*Arendt and Nübler-Jung, 1997*; *Holley et al., 1995*; *Kozmikova et al., 2013*; *Lapraz et al., 2009*; *Özüak et al., 2014*; *van der Zee et al., 2006*), it remains unclear whether this BMP-dependent body axis was a feature of the last common cnidarian–bilaterian ancestor and lost in Medusozoa, or whether it evolved independently in anthozoan Cnidaria and in Bilateria (*Genikhovich and Technau, 2017*). To address this fundamental evolutionary question, we need to gain a better understanding of how the directive axis is established and patterned.

In contrast to Bilateria, where the D-V axis and the A-P axis usually form simultaneously and very early during development, the directive body axis of the sea anemone *Nematostella vectensis* appears only at gastrula stage (*Matus et al., 2006a*; *Matus et al., 2006b*; *Rentzsch et al., 2006*), while the O-A axis is maternally determined (*Lee et al., 2007*). The expression of the core components of the BMP signaling network *bmp2/4* and *chordin* is initially controlled by β-catenin signaling and radially symmetric (*Kirillova et al., 2018*; *Kraus et al., 2016*; *Rentzsch et al., 2006*). During gastrulation, the embryo undergoes a BMP signaling-dependent symmetry break establishing the directive axis at a molecular level (*Rentzsch et al., 2006*; *Saina et al., 2009*), with a BMP signaling activity gradient forming as revealed by antibody staining against phosphorylated SMAD1/5 (pSMAD1/5) (*Figure 1C and D*; *Genikhovich et al., 2015*; *Leclère and Rentzsch, 2014*). Each end of the directive axis expresses a set of BMP ligands and BMP antagonists: *bmp2/4* and *bmp5-8* are transcribed on the low BMP signaling activity side of the directive axis together with the antagonist *chordin*, while the BMP ligand *gdf5-like* (*gdf5-l*) and the BMP antagonist *gremlin* are expressed on the high BMP signaling activity side (*Genikhovich et al., 2015*; *Rentzsch et al., 2006*). The pSMAD1/5 gradient is maintained by the genetic interactions between these molecules, with Chordin likely acting as a shuttle for BMP2/4/BMP5-8 and Gremlin serving as a primary GDF5-like antagonist (*Genikhovich et al., 2015*). Another essential player in this network is the *repulsive guidance molecule (rgm)*, which is necessary for the maintenance of the low BMP signaling side of the directive axis (*Leclère and Rentzsch, 2014*).

The graded BMP signaling activity is essential for the patterning of the endoderm and the formation of the so-called mesenteries, gastrodermal folds compartmentalizing the endoderm of the late planula into eight distinct chambers as demonstrated by knockdown (KD) experiments (*Genikhovich et al., 2015*; *Leclère and Rentzsch, 2014*). More specifically, complete abolishment of the BMP signaling gradient by KD of *bmp2/4*, *bmp5-8*, or *chordin* results in embryos which are not only molecularly but also morphologically radialized, failing to form any mesenteries (*Genikhovich et al., 2015*; *Leclère and Rentzsch, 2014*). In line with this, KD of *gdf5-l* or *gremlin* alter the profile of the BMP signaling gradient and lead to abnormal mesentery formation (*Genikhovich et al., 2015*).

Despite these important insights over the last decades, our knowledge of BMP-dependent directive axis patterning mechanisms, in particular regarding effector molecules linking BMP signaling and subsequent morphological bilaterality in *Nematostella*, remains incomplete, precluding proper comparison of anthozoan directive and bilaterian D-V axis patterning. To address this, we performed a genome-wide search for direct BMP signaling targets at two developmental stages in *Nematostella* using ChIP-seq with an anti-pSMAD1/5 antibody. We demonstrate that regulatory genes, including many with previously documented bilaterally symmetric expression, are overrepresented among direct BMP signaling targets. We also identify multiple previously uncharacterized transcription factors (TFs) and signaling molecules (SMs), whose bilaterally symmetric expression suggests that these direct BMP signaling targets may be involved in the patterning of the directive axis and in endoderm compartmentalization. Several of these seem to be shared between *Nematostella*, *Drosophila*, and *Xenopus* as shown by comparison of pSMAD1/5 ChIP-seq targets at similar developmental stages. Among the targets with maximum ChIP enrichment, we find *zswim4-6*, a gene encoding a so far uncharacterized zinc-finger protein with a SWIM domain (ZSWIM4-6), whose paralogs are also pSMAD1/5 targets in the frog. Functional analyses show that *Nematostella* ZSWIM4-6 can modulate the shape of the pSMAD1/5 gradient and appears to promote BMP signaling-mediated gene repression.

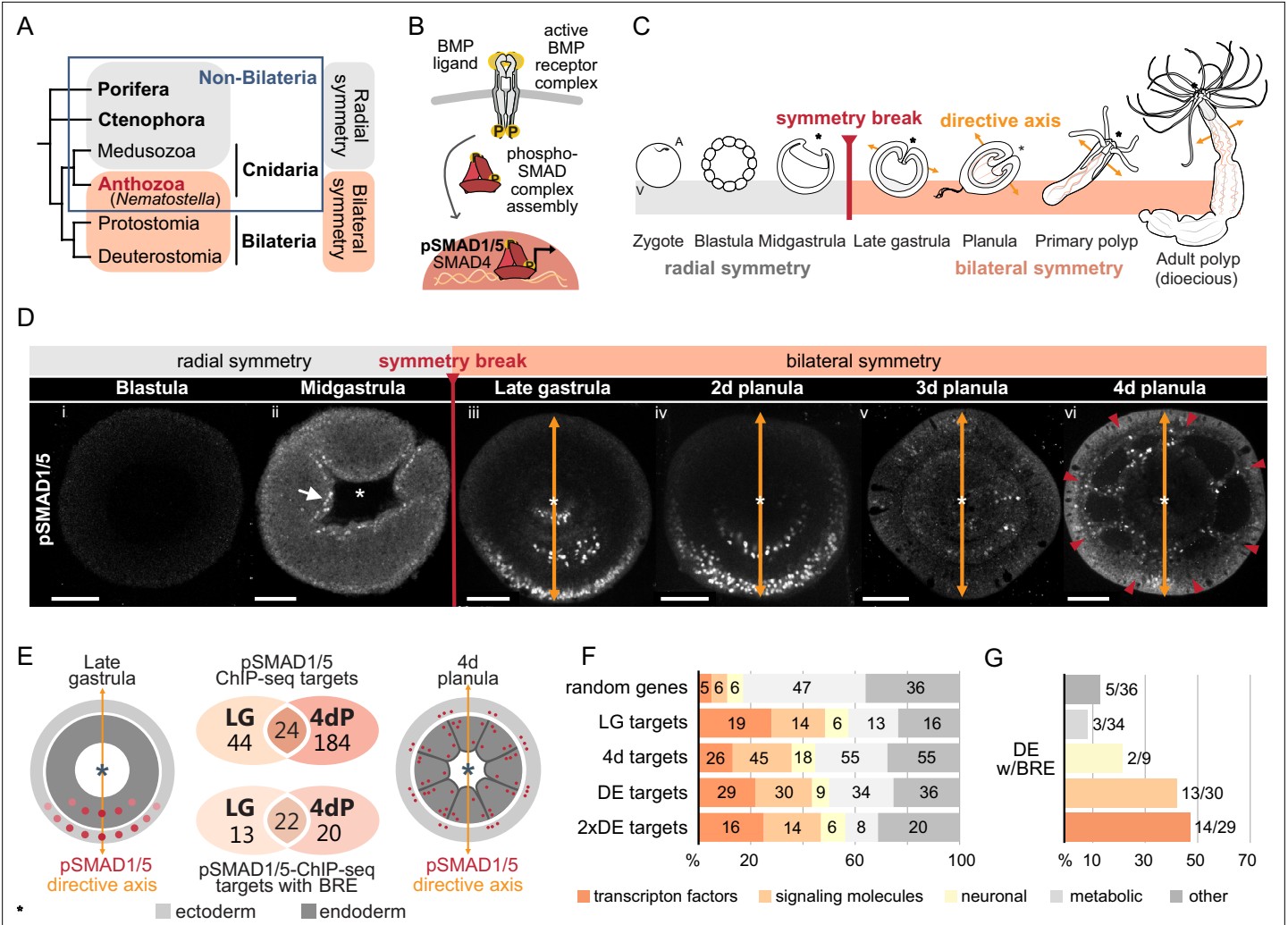

**Figure 1.** Bilateral body symmetry of the non-bilaterian sea anemone *Nematostella* is BMP signaling-dependent. (**A**) Bilateral body symmetry is observed in Bilateria and in anthozoan Cnidaria. (**B**) BMP signaling is initiated by BMP ligands binding to BMP receptors that trigger phosphorylation, assembly, and nuclear translocation of a pSMAD1/5/SMAD4 complex to regulate gene expression. (**C**) A BMP signaling-dependent symmetry break at late gastrula (LG) stage results in the formation of the secondary (directive) body axis in the sea anemone *Nematostella*. (**D**) BMP signaling dynamics during *Nematostella* development. No pSMAD1/5 is detectable in the blastula (**Di**). Nuclear pSMAD1/5 is localized in the blastopore lip of midgastrula (**Dii**), forms a gradient along the directive axis in the LG (**Diii**) and 2d planula (**Div**). By day 3, the gradient progressively disperses (**Dv**), and the signaling activity shifts to the eight forming endodermal mesenteries (**Dvi**) and to the ectodermal stripes vis-à-vis the mesenteries (arrowheads). Images (**Dii–Dvi**) show oral views (asterisks). Scale bars 50 μm. (**E**) Comparison of the direct BMP signaling targets at LG and 4dP shows little overlap. Schemes show oral views of a LG and a 4dP with red spots indicating the position of pSMAD1/5-positive nuclei in the ectoderm (light gray) and endoderm (dark gray). (**F**) Transcription factors, signaling molecules, and neuronal genes are overrepresented among the pSMAD1/5 targets compared to the functional distribution of 100 random genes. LG, late gastrula targets; 4dP, 4d planula targets; DE, pSMAD1/5 ChIP targets differentially expressed in BMP2/4 and/or GDF5-like morphants (p$_{adj}$≤0.05); 2xDE targets, pSMAD1/5 ChIP targets differentially expressed in BMP2/4 and/or GDF5-like morphants (p$_{adj}$≤0.05) showing ≥2-fold change in expression. (**G**) Fractions of each functional category of the differentially expressed pSMAD1/5 target genes (see panel **F**) containing BMP response elements (BREs).

The online version of this article includes the following figure supplement(s) for figure 1:

**Figure supplement 1.** Sequencing coverage profile shows the enrichment of pSMAD1/5 binding at the target genes.

**Figure supplement 2.** Transcriptomic comparison of BMP2/4, GDF5-like, and control morphants and differential expression of pSMAD1/5 ChIP targets upon different knockdowns.

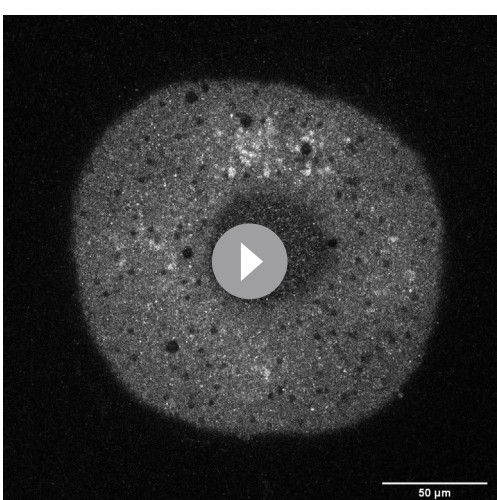

**Video 1.** BMP signaling activity in late planula. At the 4d planula larva stage, BMP signaling activity locates to the mesenteries and in the ectoderm to eight adjacent stripes that merge in pairs between the future tentacle buds to form a circumoral ring.

https://elifesciences.org/articles/80803/figures#video1

## Results

### Identification of BMP signaling targets in *Nematostella*

Staining with an antibody recognizing the transcriptionally active phosphorylated form of SMAD1/5 (pSMAD1/5) demonstrates that BMP signaling activity is highly dynamic during early development in *Nematostella*. In the early *Nematostella* gastrula, nuclear pSMAD1/5 can be detected in a radially symmetric domain around the blastopore (*Figure 1Dii*). Subsequently, it becomes restricted to one side of the gastrula and early planula (*Figure 1Diii and iv*) in a symmetry breaking process, which has been shown to depend on BMP signaling itself (*Saina et al., 2009*). The resulting BMP gradient is crucial for the formation of the directive axis. The gradient, however, disappears by late planula, when the general patterning of the directive axis is complete, and the BMP signaling activity is confined to the mesenteries, and to eight stripes following the mesenteries in the ectoderm and merging in pairs to surround the future tentacle buds and ending in a circumoral ring (*Figure 1Dv and vi*, *Video 1*).

Although the role of BMP signaling in symmetry breaking and compartmentalization of the *Nematostella* endoderm has been well described, insight regarding direct targets of BMP signaling during these fundamental developmental processes is currently lacking. To generate a genome-wide list of direct BMP signaling targets in *Nematostella*, we performed ChIP-seq using an anti-pSMAD1/5 antibody at two developmental stages: late gastrula, when BMP signaling forms a gradient along the directive axis (*Figure 1Diii*), and late planula (4d), when the mesenteries have formed and BMP signaling is confined to the mesenteries and to stripes of cells in the body wall ectoderm following the mesenteries (*Figure 1Dvi*). We found 91 significant ChIP peaks corresponding to 68 direct pSMAD1/5 target genes at late gastrula stage and 282 significant ChIP peaks corresponding to 208 pSMAD1/5 target genes in 4d planulae, 24 of which were bound by pSMAD1/5 at both developmental stages (*Figure 1E*, *Figure 1—figure supplement 1*, *Supplementary file 1*). In total, 49 out of 91 late gastrula and 67 out of the 282 4d planula ChIP peaks were located within the previously published *Nematostella* enhancers (*Schwaiger et al., 2014*). MEME-ChIP analysis (*Ma et al., 2014*) of the sequences occupied by pSMAD1/5 showed that the BMP response element (BRE) sequence, a vertebrate SMAD1/5 binding motif, was enriched at many of the bound sites. Most genes identified as pSMAD1/5 targets at both stages carry a BRE motif (22/24), with 13 additional gastrula-specific targets and 20 additional 4d planula-specific targets containing a BRE site (*Figure 1E*, *Supplementary file 1*). Next, we manually annotated pSMAD1/5 targets using reciprocal best BLAST hits and found among them many key regulatory genes whose dependence on BMP signaling has previously been documented in *Nematostella* (*Genikhovich et al., 2015*; *Leclère and Rentzsch, 2014*; *Saina et al., 2009*). These included *gbx*, *hoxB*, *hoxD,* and *hoxE,* that is, genes shown to play a central role in the subdivision of the directive axis into mesenterial chambers and specifying the fate of each mesenterial chamber (*He et al., 2018*), as well as the BMP signaling regulators *chordin*, *gremlin,* and *rgm* (*Genikhovich et al., 2015*; *Leclère and Rentzsch, 2014*; *Saina et al., 2009*). Intriguingly, none of the BMP ligand-coding genes, that is, *bmp2/4*, *bmp5-8,* or *gdf5-like*, were found among the direct BMP signaling targets (*Supplementary file 1*).

To expand this analysis and gain insight into the response of the 252 newly found putative direct pSMAD1/5 target genes to abolished or reduced BMP signaling, we injected previously characterized antisense morpholino oligonucleotides against *bmp2/4* and *gdf5-like* (*Genikhovich et al., 2015*; *Saina et al., 2009*), as well as a standard control morpholino, and analyzed by RNA-seq whether the pSMAD1/5 target genes were differentially expressed upon *gdf5-like* or *bmp2/4* KD (*Figure 1—figure*

*supplement 2*, *Supplementary file 1*). In total, 139 direct pSMAD1/5 target genes were among the genes differentially expressed in the KD ($p_{adj}$≤0.05), 64 of which showed more than a twofold change in expression upon BMP2/4 and/or GDF5-like KD (*Supplementary file 1*). The transcriptional response of the previously described downstream targets of BMP signaling confirmed previous in situ hybridization analysis, that is, *chordin* and *rgm* were upregulated while *gbx*, *hox* genes and *gremlin* were downregulated in our RNA-seq datasets confirming their specificity. Notably, out of 55 pSMAD1/5 target genes with BRE motifs, 37 (67.3%) were shown to be differentially expressed upon BMP2/4 and/or GDF5-like KD (*Supplementary file 1*).

To gain insight into the function of pSMAD1/5 target genes in *Nematostella*, we put them into broad categories according to the putative function of their bilaterian homologs. We found that the fraction of TFs and SMs was strongly increased among the direct BMP signaling targets at both developmental stages in comparison to 100 randomly selected genes from the *Nematostella* genome (*Figure 1F and G*). This suggests that direct downstream targets of BMP signaling constitute a second tier of a regulatory cascade governing patterning and morphogenesis of the *Nematostella* gastrula and planula.

Since we were particularly interested in identifying new players in the BMP-dependent regulation of the directive axis, we characterized the expression domain of a subset of pSMAD1/5 targets with so far unknown expression patterns and validated the expression of several previously described pSMAD1/5 targets using whole mount in situ hybridization. In accordance with a potential role in the directive axis regulation, we found several of the newly identified pSMAD1/5 targets to be bilaterally expressed (e.g., *otxB*, *tbx2/3*, *tbx20.1*, *p63*, *dusp1*, *bmprII*, *c-ski*, *morn*, *pik*, *zswim4-6*) (*Figure 2*), further motivating the investigation of their function in directive axis patterning.

## Possible conservation of direct BMP signaling targets between Anthozoa and Bilateria

To identify BMP signaling targets conserved between Anthozoa and Bilateria, we compared *Nematostella* pSMAD1/5 targets with available ChIP-seq data from two bilaterians (*Figure 3A*). In *Drosophila melanogaster*, pMAD targets were identified at two developmental time points (2 and 3 hr after fertilization [hpf]) (*Deignan et al., 2016*), and in *Xenopus laevis*, SMAD1 targets were found postgastrulation (NF20 stage) (*Stevens et al., 2017*). In a three-way comparison of *Nematostella* (*Nve*), fly (*Dme*), and frog (*Xla*), putative orthologs were identified using best reciprocal BLAST hits based on the bit score. We found 103 direct BMP signaling targets that were conserved between at least two organisms, of which four were shared by all three species (*Figure 3B and C*). Analyzing the targets in *Nematostella* with two-way reciprocal best BLAST hits to targets in *Drosophila* or *Xenopus* suggests that a total of at least eight can be assigned to all three species (*meis*, *gata*, *tbx2/3*, *nkain*, *irx*, *zfp36l1*, *ptc*, *tp53bp2*). Among the shared targets, conserved TFs and SMs are enriched compared to genes with other functions (69/103). Multiple Hox genes are direct targets of BMP signaling in embryos of *Nematostella* (*hoxB*, *hoxD*, *hoxE*), fly (*dfd*, *pb*, *antp*), and frog (*hoxa1-7/9-11/13*, *hoxb1-9*, *hoxc3-6/8-12*, *hoxd1/3/4/8-11/13*); however, the orthology of the cnidarian and bilaterian Hox genes is unclear due to the likely independent diversification of the 'anterior' and 'non-anterior' Hox genes in these two sister clades (*Chourrout et al., 2006*; *Genikhovich and Technau, 2017*).

## *zswim4-6:* A previously uncharacterized, asymmetrically expressed gene encoding a nuclear protein

One of the highest enrichment levels in our ChIP-seq was detected for *zswim4-6* – a pSMAD1/5 target shared between *Nematostella* and *Xenopus* (*Figure 3C*). This gene encodes a zinc-finger containing protein with a SWIM-domain clustering together with ZSWIM4/5/6 of zebrafish (*Danio rerio*), frog (*Xenopus tropicalis*), and mouse (*Mus musculus*) (*Figure 4A*). *Zswim4-6* starts to be expressed at the early gastrula stage around the blastopore (*Figure 4B–C'*), concomitant with the onset of BMP signaling activity. At late gastrula, when a symmetry break in the BMP signaling activity establishes the directive axis, *zswim4-6* expression becomes restricted to the ectoderm and endoderm on one side of it (*Figure 4D and D'*). Double in situ hybridization of *chd* and *zswim4-6* shows that *zswim4-6* is expressed opposite to *chd*, that is, on the side of high pSMAD1/5 activity (*Figure 4E and E'*), suggesting that BMP signaling activates *zswim4-6* expression. At late planula stage, strong *zswim4-6* expression is observed in the eight mesenteries (*Figure 4F and F'*). Taken together, in situ hybridization

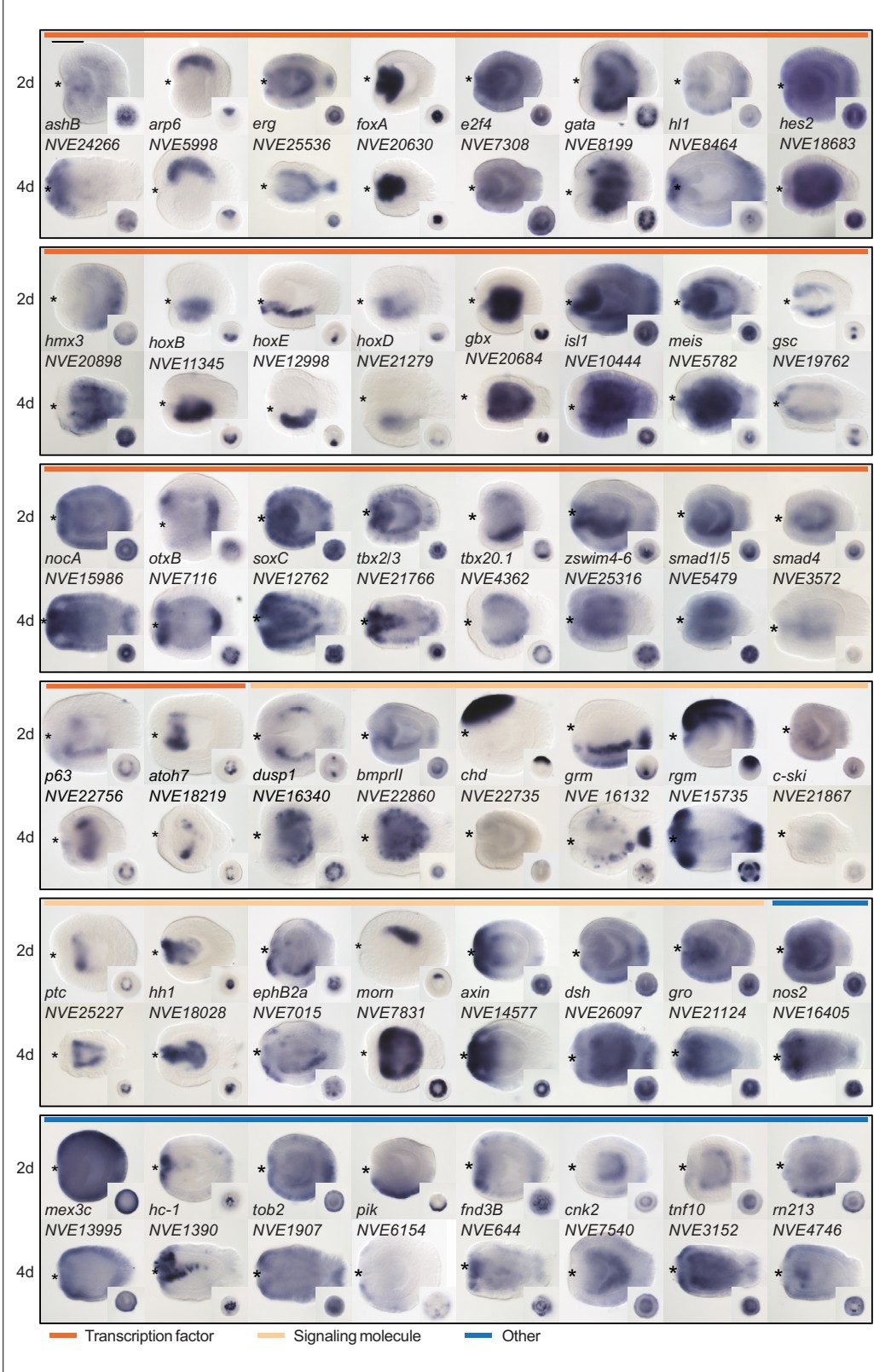

**Figure 2.** Expression patterns of a selection of the direct targets of BMP signaling in 2d and 4d planulae. In lateral views, the oral end is marked with an asterisk, inlets show oral views. Scale bar 100 μm.

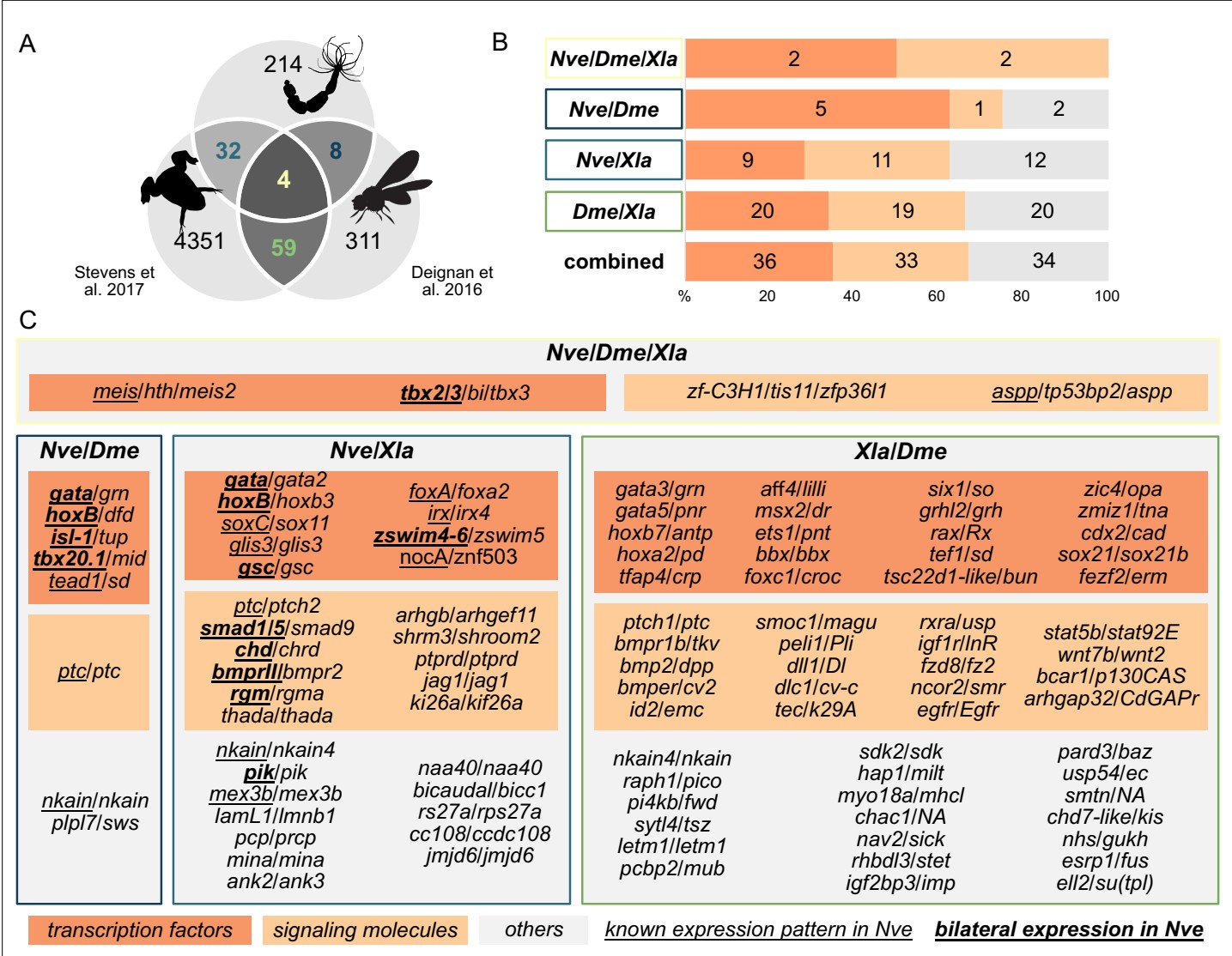

**Figure 3.** *Nematostella* and Bilateria share BMP signaling targets, which predominantly encode transcription factors and signaling molecules. (**A**) Overlap of BMP signaling targets at comparable embryonic stages in the three-way comparison of *Nematostella*, *Drosophila* (*Deignan et al., 2016*), and *Xenopus* (*Stevens et al., 2017*). Orthology links were deduced by NCBI BLASTP of the respective proteomes with a cut-off e-value ≤ 1e-5, and reciprocal best BLAST hits were determined using the bit score. (**B**) Transcription factors and signaling molecules represent more than 60% of pSMAD1/5 targets shared between *Nematostella* (*Nve*), fly (*Dme*), and frog (*Xla*). (**C**) Gene names of orthologous targets shared between *Nematostella*, fly, and frog. For targets shared with *Nematostella*, genes with known expression patterns in the embryo are underlined, while genes expressed asymmetrically along the directive axis are underlined and bold.

analysis showed that the *zswim4-6* expression domain followed the dynamic changes in pSMAD1/5 activity (compare *Figure 1D* and *Figure 4B–F'*).

Sequence analysis of the deduced ZSWIM4-6 protein revealed a potential N-terminal nuclear localization signal (NLS). To analyze the intracellular localization of ZSWIM4-6, we mosaically overexpressed an EGFP-tagged version of ZSWIM4-6 in F0 transgenic animals under the control of the ubiquitously active *EF1α* promoter (*Kraus et al., 2016*; *Steinmetz et al., 2017*). We compared the intracellular localization of the tagged wild-type form to a truncated version of ZSWIM4-6 (ZSWIM4-6ΔNLS-EGFP), where the first 24 bases coding for the putative NLS were replaced by a start codon. Full-length ZSWIM4-6-EGFP was enriched in the cell nuclei (*Figure 4G*), while the truncated version of ZSWIM4-6 lacking the NLS remained cytoplasmic, similar to the EGFP signal in transgenic animals overexpressing only cytoplasmic EGFP (*Figure 4H and I*). Nuclear localization of ZSWIM4-6 could be confirmed by microinjection of *zswim4-6-EGFP* mRNA (*Figure 4J–L*). Strikingly, the nuclear fluorescent signal of

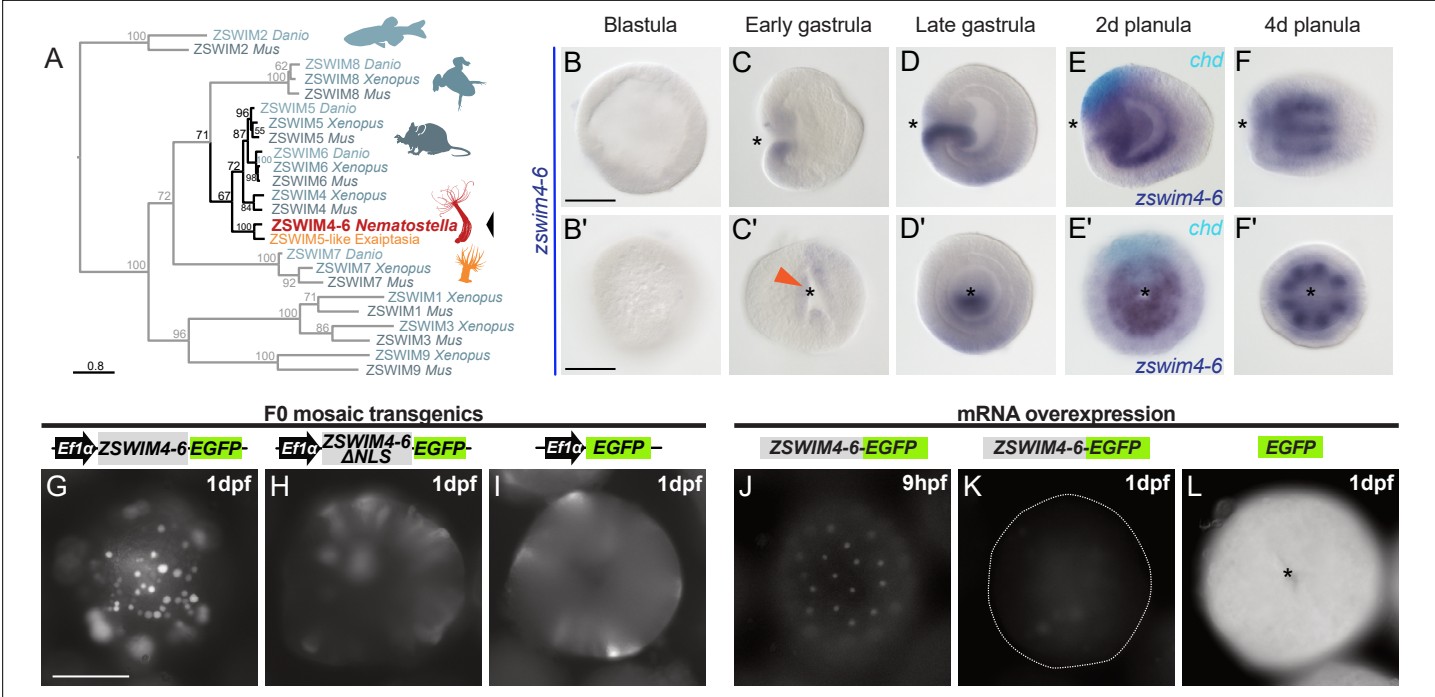

**Figure 4.** *zswim4-6* is a target of BMP signaling with bilaterally symmetric expression and encodes a nuclear protein. (**A**) Maximum likelihood phylogeny shows that *Nematostella* ZSWIM4-6 clusters with ZSWIM4, ZSWIM5, and ZSWIM6 from zebrafish, frog, and mouse. (**B–F'**) *Nematostella zswim4-6* expression follows the dynamic BMP signaling domain (see *Figure 1D* for comparison). Double ISH shows *zswim4-6* and *chd* transcripts localize to the opposite sides of the directive axis. (**G–I**) Mosaic expression of ZSWIM4-6-EGFP under the control of the ubiquitously active *EF1α* promoter in F0 transgenic animals demonstrates that ZSWIM4-6 is a nuclear protein. Full-length ZSWIM4-6-EGFP is translocated into the nuclei (**G**), while ZSWIM4-6ΔNLS-EGFP missing the predicted nuclear localization signal NLS remains cytoplasmic (**H**), similar to the EGFP control (**I**). Exposure time was the same in all images. (**J–L**) Microinjection of *ZSWIM4-6-EGFP* mRNA results in a weak EGFP signal detectable in the nuclei of the early blastula (**J**), which progressively disappears towards late gastrula (**K**). EGFP translated from *EGFP* mRNA remains readily detectable (**L**). To visualize the weak signal in (**J–K**), the exposure had to be increased in comparison to (**L**). Asterisks mark the oral side; scale bars 100 μm.

ZSWIM4-6-EGFP, which is weak but clearly visible at the early blastula stage (*Figure 4J*), progressively decreases and is barely detectable at 24 hpf (*Figure 4K*). In contrast, injection of mRNA encoding cytoplasmic EGFP results in a strong fluorescent signal lasting for several days (*Figure 4L*). Currently, the dynamics of the endogenous ZSWIM4-6 protein, as well as the mechanism behind this fast turn-over after overexpression, are unclear.

To analyze the function of BMP signaling in establishing the *zswim4-6* expression domain, we performed KDs of several components of the BMP signaling network (*Figure 5A*). Previous studies have shown that the KDs of *bmp2/4* and *chd* suppress BMP signaling activity and abolish the pSMAD1/5 gradient (*Genikhovich et al., 2015*; *Leclère and Rentzsch, 2014*). A KD of another BMP ligand, GDF5-like, reduces BMP signaling activity, although a shallow BMP signaling gradient is preserved, while the KD of the BMP inhibitor Gremlin results in an expansion of the nuclear pSMAD1/5 gradient (*Genikhovich et al., 2015*; *He et al., 2023*). In line with the proposed role of BMP signaling in activating *zswim4-6* expression, the KDs of *bmp2/4* and *chd* led to a loss of *zswim4-6* expression (*Figure 5B–D'*, *Supplementary file 1*). In contrast, KD of *gdf5-like* and *gremlin* did not strongly affect *zswim4-6* expression (*Figure 5E and F'*, *Supplementary file 1*), suggesting that *zswim4-6* expression might either specifically depend on the BMP2/4-mediated, but not GDF5-like-mediated BMP signaling, or be robust toward milder alterations in pSMAD1/5 levels, or both.

BMP ligand-specific activation of *zswim4-6* would require differential selectivity of the BMP receptors toward BMP2/4 or GDF5-like (similar to the situation described in *Heinecke et al., 2009*; *Nickel and Mueller, 2019*). Therefore, we analyzed the expression of *zswim4-6* upon suppression of each of the two *Nematostella* BMP type I receptors (*alk2* and *alk3/6*), and a single BMP type II receptor (BMPRII) (*Figure 5—figure supplement 1*). We suppressed Alk2, Alk3/6, and BMPRII either by microinjecting mRNA of the dominant-negative version of each of these receptors, in which the intracellular

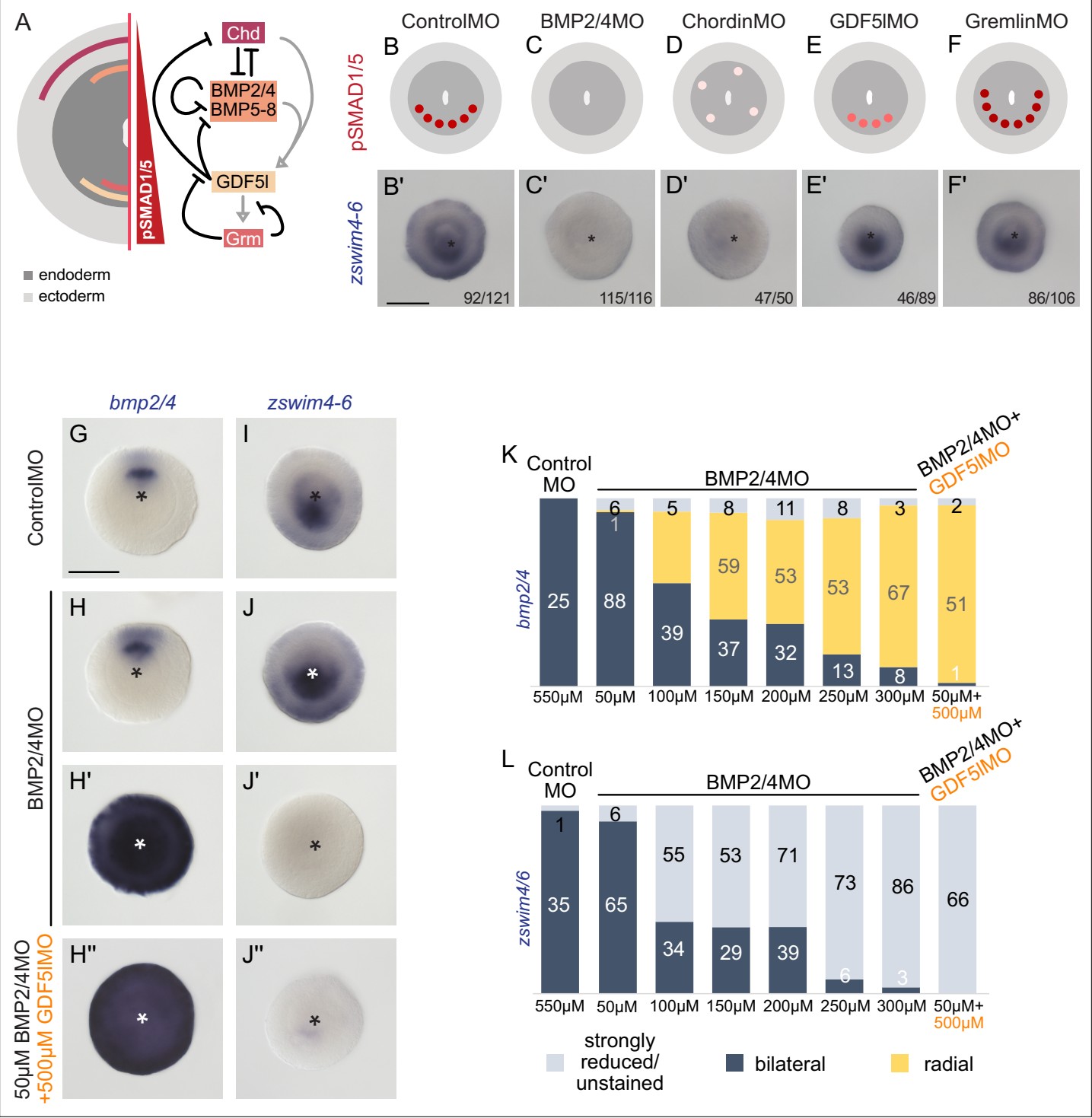

**Figure 5.** *zswim4-6* reacts differently to the downregulation of BMP2/4 and GDF5-like. (**A**) Expression domains and putative interactions of the BMP network components in 2d embryos (based on *Genikhovich et al., 2015*). (**B–F'**) Effect of the knockdown (KD) of the individual BMP signaling components on nuclear pSMAD1/5 and on *zswim4-6* expression. *zswim4-6* expression is abolished upon KD of *bmp2/4* and *chordin*, but not in KD of *gdf5-l* and *gremlin*. Sketches in (**B–F**) show that the gradient is lost upon BMP2/4 and Chd KD, reduced upon GDF5-like KD and expanded upon Gremlin KD (based on *Genikhovich et al., 2015*). (**G–L**) KD of *bmp2/4* using different concentrations of BMP2/4MO results in either normal, bilaterally symmetric marker expression or complete radialization but no intermediate phenotypes. Instead, the penetrance of the radialization phenotype increases with the increase of the BMP2/4MO concentration. Co-injection of the lowest, ineffective BMP2/4MO concentration with GDF5lMO also results in a complete radialization. All stainings are performed on 2d planulae. Scale bars 100 μm.

*Figure 5 continued on next page*

*Figure 5 continued*

The online version of this article includes the following source data and figure supplement(s) for figure 5:

**Figure supplement 1.** Two type I BMP receptors display partial redundancy in regulating *zswim4-6* expression.

**Figure supplement 1—source data 1.** Source data for *Figure 5—figure supplement 1D*.

kinase domain was replaced by EGFP, or by shRNA-mediated RNAi (*Figure 5—figure supplement 1*). In both assays, *zswim4-6* expression appears to be stronger affected by the Alk3/6 suppression than by the Alk2 suppression and is nearly abolished by the combined Alk3/6/Alk2 KD (*Figure 5—figure supplement 1*). In contrast, it is largely unaffected by BMP type II receptor suppression (*Figure 5—figure supplement 1*), suggesting that BMPRII KD during early *Nematostella* development can be compensated for – likely by another type II TGFβ receptor molecule. Importantly, however, neither Alk2 nor Alk6 proved to be solely responsible for the BMP-dependent *zswim4-6* expression, and only their simultaneous KD led to a strong suppression of *zswim4-6*, which speaks against the hypothesis that Alk2 and Alk3/6 are strictly selective toward specific BMP ligands.

Next, we tested the response of *zswim4-6* to reduced levels of BMP signaling. We injected *Nematostella* embryos with different concentrations of BMP2/4MO starting with 50 µM and going up to the regular BMP2/4MO working concentration of 300 µM in steps of 50 µM, and stained these embryos with probes against *zswim4-6*, a positively regulated gene, and *bmp2/4*, whose expression is suppressed by BMP signaling. Strikingly, within individual embryos, the response to the different concentrations of the BMP2/4MO was binary: either *bmp2/4* and *zswim4-6* expression appeared normal (*Figure 5H–J*), or it was completely radialized or lost for, respectively, *bmp2-4* (*Figure 5H'*) and *zswim4-6* (*Figure 5J'*). Intermediate phenotypes were never observed. However, the proportion of the embryos showing radialized *bmp2/4* and abolished *zswim4-6* expression increased depending on the amount of injected BMP2/4MO (*Figure 5K and L*). This is in line with an earlier observation of the binary effect of injecting iteratively decreasing concentrations of BMP2/4MO on mesentery formation (*Leclère and Rentzsch, 2014*). These results suggest that alterations in the BMP2/4-mediated signaling may not change the shape of the BMP signaling gradient, to which *zswim4-6* expression could be more or less robust, but rather define whether there is a BMP signaling gradient or not. Since the binary effect of the injection of different concentrations of the BMP2/4MO prevented us from gradually reducing the BMP signaling, these experiments did not exclude the possibility that the low level of nuclear pSMAD1/5 observed upon *gdf5l* KD (*Genikhovich et al., 2015*) was sufficient to activate the *zswim4-6* expression. To test this, we co-injected 50 µM BMP2/4MO, which was too low a concentration to affect *zswim4-6* expression (*Figure 5L*), together with 500 µM GDF5lMO, which alone also did not significantly affect *zswim4-6* expression (*Figure 5E'*). Strikingly, when co-injected, these two MOs phenocopied the effect of the 300 µM BMP2/4MO, completely radializing the *bmp2/4* expression and abolishing *zswim4-6* (*Figure 5H"–J" and K–L*). Thus, although BMP2/4-specific *zswim4-6* activation cannot be entirely ruled out by this double-KD experiment (*Figure 5H" and J"*), it currently seems likely that *zswim4-6* expression is not BMP ligand-specific but rather appears to be activated even by the slightest BMP input.

## ZSWIM4-6 is a modulator of BMP-dependent patterning

Given that *zswim4-6* expression is bilaterally symmetric and highly dependent on BMP signaling in *Nematostella*, we wanted to see whether *zswim4-6* itself contributes to the patterning of the directive axis. To this end, we performed KD of *zswim4-6* activity using a translation-blocking morpholino targeting *zswim4-6* (*Figure 6—figure supplement 1*).

Analysis of the overall morphology at late planula stage showed that *zswim4-6* KD leads to defects in endoderm compartmentalization. In control morphants, eight mesenteries reached all the way to the pharynx, and the primary polyp had four tentacles (*Figure 6A–A"*). In contrast, in ZSWIM4-6MO, the mesenterial chamber expressing HoxE protein failed to reach the pharynx, resulting in the fusion of the two neighboring mesenterial segments (*Figure 6B and B'*). Consequently, *zswim4-6* morphant primary polyps formed only three instead of the typical four tentacles (*Figure 6B"*). Endoderm segmentation defects have previously been observed in *Nematostella* larvae with defects in pSMAD1/5 gradient formation, as well as in *Hox* gene activity (*Genikhovich et al., 2015*; *He et al., 2018*; *Leclère and Rentzsch, 2014*), and, based on its expression domain, *zswim4-6* may be involved

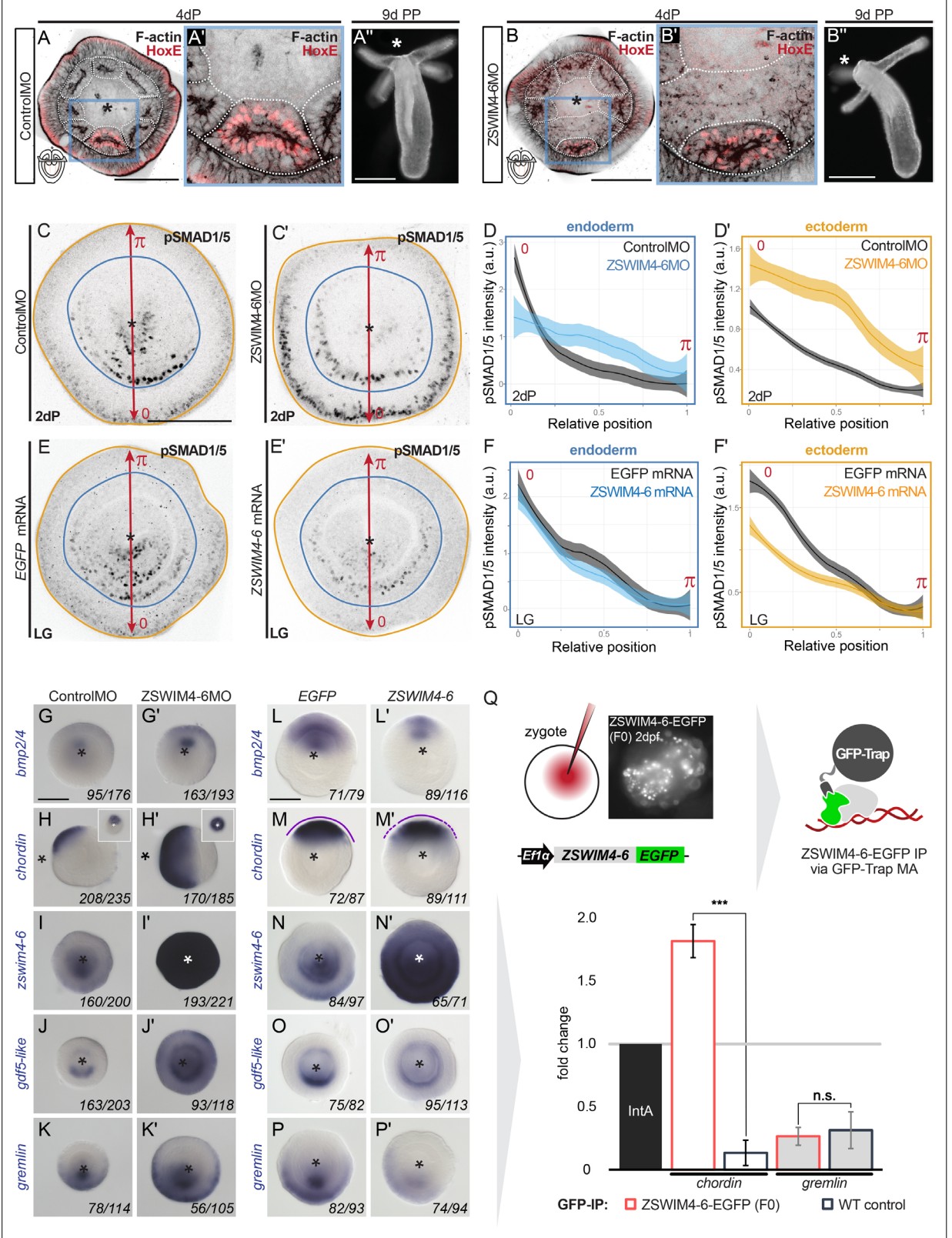

**Figure 6.** ZSWIM4-6 is a modulator of BMP signaling that appears to act as transcriptional repressor. (**A–B"**) Morpholino knockdown (KD) of *zswim4-6* results in patterning defects. (**A**) and (**B**) show confocal sections across the pharyngeal region of 4d planulae stained with antiHoxE antibody (*Genikhovich et al., 2015*) and phalloidin. (**A'**) and (**B'**) show the areas boxed in (**A**) and (**B**). White dotted lines delineate mesenterial chambers. In the 4d planula, the HoxE-positive mesenterial chamber does not reach the pharynx, which leads to the fusion of neighboring chambers (compare **A**, **A'** with

*Figure 6 continued*

**B**, **B'**). This results in the formation of three instead of four tentacles in the 9d polyp (compare **A"** with **B"**). (**C–F'**) Immunofluorescence and quantification of relative nuclear anti-pSMAD1/5 staining intensities in 2d ZSWIM4-6 morphants (**C–D'**) and upon *zswim4-6* mRNA overexpression in the late gastrula (**E–F'**). Intensity measurements (arbitrary units, a.u.) are plotted as a function of the relative position of each nucleus in the endoderm or in the ectoderm along a 180° arc from 0 (high signaling side) to π (low signaling side). The measurements from Control MO embryos (n = 10) and ZSWIM4-6MO embryos (n = 10), as well as *egfp* mRNA embryos (n = 22 for the endodermal, and n = 24 for the ectodermal measurements) and *zswim4-6* mRNA embryos (n = 8 for the endodermal, and n = 9 for the ectodermal measurements) are described by a LOESS smoothed curve (solid line) with a 99% confidence interval for the mean (shade). For visualization purposes, the intensity values were normalized to the upper quantile value among all replicates and conditions of each control-experiment pair. (**G–K'**) Expression of *zswim4-6* and BMP network components in the 2d planula upon morpholino KD of *zswim4-6*; All images except for (**H**) and (**H'**) show oral views. (**L–P'**) Expression of *zswim4-6* and BMP network components in late gastrula (30 hr) upon *zswim4-6* mRNA injection; oral views, purple dashed line marks the loss of a sharp boundary of *chd* expression. In (**A–P'**), asterisks mark the oral side. In (**A-P'**), scale bars 100 µm. (**Q**) ChIP with GFP-Trap detects ZSWIM4-6-EGFP fusion protein in the vicinity of the pSMAD1/5 binding site in the upstream regulatory region of *chordin* but not of *gremlin*. Experiment on biological quadruplicates. Mean enrichments and standard deviations are shown.

The online version of this article includes the following source data and figure supplement(s) for figure 6:

**Source data 1.** Source data for *Figure 6D-F'*.

**Source data 2.** Source data for *Figure 6Q*.

**Figure supplement 1.** Testing ZSWIM4/6 morpholino specificity.

**Figure supplement 2.** *zzswim5* overexpression dampens BMP signaling and causes developmental defects in zebrafish.

**Figure supplement 2—source data 1.** Source data for *Figure 6—figure supplement 2G*.

**Figure supplement 3.** ZSWIM4-6 knockdown partially rescues the reduction of pSMAD1/5 caused by GDF5-like knockdown.

**Figure supplement 3—source data 1.** Source data for *Figure 6—figure supplement 3E and F*.

**Figure supplement 4.** *rgm* expression expands upon ZSWIM4-6 knockdown.

**Figure supplement 5.** CRISPR/Cas9-mediated mutagenesis of *zswim4-6* results in the expansion of *chordin* expression in mosaic mutants (F0).

in either of these two processes. However, in contrast to the *HoxE* mutants, which also have characteristic three-tentacle primary polyps due to the loss of the *HoxE*-expressing mesenterial chamber and the fusion of the *HoxD*-expressing mesenterial chambers capable of forming a tentacle (*He et al., 2018*), the affected pair of mesenteries is not lost in the *zswim4-6* morphants, and HoxE is detectable (*Figure 6B'*). Therefore, we concentrated on analyzing the molecular role of *zswim4-6* in the BMP signaling network.

First, we assessed whether the pSMAD1/5 gradient was affected upon *zswim4-6* KD. To this end, we performed anti-pSMAD1/5 immunostainings in 2d planulae and quantified pSMAD1/5 levels along the directive axis. We found that the gradient profile of pSMAD1/5 in the endoderm was flattened in *zswim4-6* morphants compared to controls, and that peak levels of pSMAD1/5 activity were not reached. pSMAD1/5 activity in the ectoderm was, however, strongly elevated upon *zswim4-6* KD (*Figure 6C–C' and D–D'*). To corroborate this, we analyzed the pSMAD1/5 gradient in embryos microinjected with *zswim4-6-EGFP* mRNA. Due to the very fast turnover of ZSWIM4-6-EGFP (*Figure 4J–L*), we quantified the gradient at the earliest possible stage after the symmetry break, that is, at late gastrula. *zswim4-6* mRNA overexpression did not result in significant changes in the shape of the pSMAD1/5 gradient in the endoderm; however, ectodermal pSMAD1/5 levels were significantly reduced in the area where BMP signaling is normally strongest (*Figure 6E–E' and F–F'*). The expansion of the pSMAD1/5 domain upon *Nematostella zswim4-6* KD and its reduction upon *zswim4-6* overexpression suggests that SMAD1/5 may be stabilized in the absence of ZSWIM4-6. This is in line with the effect of the overexpression of zebrafish *zzswim5* on the pSmad1/5/9 gradient in zebrafish embryos (*Figure 6—figure supplement 2*), as well as with the recently reported role of the vertebrate *Zswim4* in frog embryos and mammalian cells (*Wang et al., 2022*). There, it has been shown that Zswim4 attenuates BMP signaling by directly interacting with nuclear SMAD1 and promoting its ubiquitination and degradation (*Wang et al., 2022*). To test whether *Nematostella* zswim4-6 may play a role in destabilizing nuclear SMAD1/5, we analyzed the shape of the nuclear pSMAD1/5 gradient in the GDF5-like morphants co-injected either with Control MO or with ZSWIM4-6MO. As we showed previously (*Genikhovich et al., 2015*), pSMAD1/5 staining was drastically reduced upon GDF5lMO injection, but adding ZSWIM4-6MO to the GDF5lMO indeed led to a much milder reduction of the pSMAD1/5 gradient (*Figure 6—figure supplement 3*), supporting the ZSWIM4-6-mediated pSMAD1/5 destabilization hypothesis.

We then evaluated how KD of *zswim4-6* affected the transcription of markers expressed on the low BMP and high BMP signaling ends of the directive axis. First, we looked at *bmp2/4, chd,* and *rgm* – the key regulators of pSMAD1/5 gradient formation in *Nematostella,* whose expression is repressed by high levels of BMP signaling (*Genikhovich et al., 2015*; *Leclère and Rentzsch, 2014*; *Saina et al., 2009*). The *bmp2/4, chd,* and *rgm* expression domains were expanded in *zswim4-6* morphants compared to controls (*Figure 6G–H'*, *Figure 6—figure supplement 4*), which was especially evident in the ectodermal expansion of *chd* and *rgm*. Similarly, the *chd* expression domain was also expanded in the mosaic F0 *zswim4-6* knockouts generated by CRISPR/Cas9, corroborating the morpholino KD results (*Figure 6—figure supplement 5*). This was striking and unexpected since *bmp2/4, chd,* and *rgm* are repressed by BMP signaling, and their expansion was in spite of the much stronger pSMAD1/5 signal observed in the ectoderm of the ZSWIM4-6 morphants (*Figure 6C–D'*). Next, we looked at the markers of the high BMP signaling end: *zswim4-6, gdf5-like,* and *grm,* which are positively regulated by BMP signaling. The KD of *zswim4-6* translation resulted in a strong upregulation of *zswim4-6* transcription, especially in the ectoderm, suggesting that ZSWIM4-6 might either act as its own transcriptional repressor or that *zswim4-6* transcription reacts to the increased ectodermal pSMAD1/5 (*Figure 6I and I'*). The expression domain of *gdf5-like* (*Figure 6J and J'*) and *grm* (*Figure 6K and K'*) was expanded in the ectoderm, which reflects the flattening of the endodermal and the expansion of the ectodermal pSMAD1/5 gradient in the morphants (*Figure 6C–D'*).

The overexpression of *zswim4-6-EGFP* by mRNA injection resulted in largely opposite effects on gene expression compared to the morpholino-mediated KD of ZSWIM4-6, although less pronounced. As expected from the analysis of the morphants, *zswim4-6-egfp* overexpression caused a mild reduction of the expression domains of *bmp2/4* and *chd* (*Figure 6L–M'*). Moreover, the otherwise very sharp border of the *chd* expression domain appeared diffuse (*Figure 6M'*). In situ hybridization with the *zswim4-6* probe detected exogenous *zswim4-6* throughout the injected embryo (*Figure 6N and N'*). *gdf5-like* expression (*Figure 6O and O'*) appeared reduced on the 'strong gdf5-like side', and *grm* was reduced as well (*Figure 6P and P'*), which also reflects the behavior of the pSMAD1/5 gradient (*Figure 6F and F'*). Taken together, in the absence of ZSWIM4-6, changes in the expression of genes expressed on the high pSMAD1/5 side (*gdf5-like, grm*) appear to mimic the changes in the levels and range of the BMP signaling gradient. In contrast, genes repressed by BMP signaling and expressed on the low BMP signaling side of the directive axis (*bmp2/4, chd*) seem to expand their expression domains in the absence of ZSWIM4-6 despite the increase in pSMAD1/5 levels. One possible explanation for this observation is that BMP signaling-mediated gene repression is less effective in the absence of *ZSWIM4-6*.

To test this hypothesis, we wanted to analyze whether ZSWIM4-6 was differentially bound to pSMAD1/5 binding sites at the genes repressed by BMP signaling and at the genes activated by BMP signaling. Due to the lack of an anti-ZSWIM4-6 antibody and fast ZSWIM4-6-EGFP turnover in mRNA-injected embryos (*Figure 4J and K*), we microinjected *EF1α::ZSWIM4-6-EGFP* plasmid (*Figure 4G*) to perform anti-GFP-ChIP on mosaic F0 embryos, where *zswim4-6-egfp* is transcribed under control of a ubiquitously active promoter in a small fraction of cells in each embryo. Since the amount of chromatin that could be immunoprecipitated in such an experimental setup was very low, we were limited to selecting one negatively regulated direct BMP target and one positively regulated direct BMP target for the subsequent qPCR analysis. We chose pSMAD1/5 sites in the regulatory regions of the negatively regulated direct BMP target *chordin* and the positively regulated direct BMP target *gremlin,* and checked by qPCR whether they were bound by ZSWIM4-6-EGFP. Binding to the *Intergenic region IntA* (*Schwaiger et al., 2021*; *Schwaiger et al., 2014*) was used as a normalization control. The pSMAD1/5 binding site-containing region at the *gremlin* locus was depleted in both wild type and in *EF1α::ZSWIM4-6-EGFP*-injected embryos upon αGFP-ChIP (0.35-fold enrichment and 0.27-fold enrichment, respectively; not significant according to the two-tailed *t*-test; p=0.496). On the other hand, the pSMAD1/5 binding site-containing region at the *chordin* locus was 1.82-fold enriched upon anti-GFP-ChIP in *EF1α::ZSWIM4-6-EGFP* embryos and 0.15-fold enriched (significant according to the two-tailed *t*-test; p=0.00032) in wild-type embryos (*Figure 6Q*). Thus, the enrichment of the pSMAD1/5 binding site-containing region at the *chordin* locus in *EF1α::ZSWIM4-6-EGFP* embryos in relation to wild-type embryos was 12.13-fold, in stark contrast to the 0.87-fold enrichment at the *gremlin* locus in *EF1α::ZSWIM4-6-EGFP* embryos in comparison to the wild-type embryos. This suggests an interaction of ZSWIM4-6 with the pSMAD1/5 binding site of the *chordin* regulatory

region, but not of the *gremlin* regulatory region, and supports the hypothesis that ZSWIM4-6 might act as a co-repressor for pSMAD1/5 targets.

## Discussion

### BMP signaling governs the expression of a second tier of developmental regulators

BMP signaling regulates the patterning of the anthozoan directive axis and the bilaterian D-V axis; however, the extent of conservation of downstream target genes between Bilateria and Cnidaria involved in the axial patterning has been largely unknown. In our anti-pSMAD1/5-ChIP-seq in *Nematostella*, we identified putative direct targets of BMP signaling during the establishment of the directive axis in the late gastrula and in the fully compartmentalized 4d planula, providing new insights into the genetic program responsible for directive axis patterning. Functional annotation of the identified targets highlights an abundance of TFs and SMs (*Figure 1F*), suggesting that BMP signaling regulates a second tier of regulators rather than structural genes at the two stages we assayed. TFs and SMs were also prominent among the differentially expressed target genes upon KD of the BMP ligands BMP2/4 and GDF5-like (*Figure 1F*) and among the targets with an identifiable BRE (*Katagiri et al., 2002*). The presence of multiple members of Wnt, MAPK, Hedgehog, and Notch signaling pathways among the BMP signaling targets points at a high degree of coordination between different signal transduction cascades necessary to generate a properly organized embryo. Comparison of the *Nematostella* pSMAD1/5 targets with the available ChIP-seq data of the frog *Xenopus* (*Stevens et al., 2017*) and the fly *Drosophila* (*Deignan et al., 2016*) shows that BMP targets shared between Anthozoa and Bilateria are also mainly TFs and SMs.

Several molecules of the BMP pathway in Bilateria are known to be directly regulated by BMP signaling (*Figure 3C*; *Deignan et al., 2016*; *Genander et al., 2014*; *Greenfeld et al., 2021*; *Rogers et al., 2020*; *Stevens et al., 2017*). Likewise, pSMAD1/5 in *Nematostella* directly controls the transcription of multiple BMP network components (*chd, grm, bmprII, rgm, smad1/5, smad4*) and other associated regulators (*e2f4, morn, tob2, c-ski, tld1-like, bmp1-like, dusp1, dusp7*) that have not been characterized yet. Curiously, unlike *Xenopus bmp2, -4, -5, -7*, and *gdf2, -6, -9*, and *-10*, or *Drosophila dpp* (*Deignan et al., 2016*; *Stevens et al., 2017*), the promoters of *Nematostella* BMP-encoding genes do not appear to be directly bound by pSMAD1/5. Our ChIP-seq data suggest that BMP signaling-dependent repression of *Nematostella bmp2/4* and *bmp5-8*, and BMP signaling-dependent activation of *gdf5-like* (*Genikhovich et al., 2015*) is indirect, but the link between the pSMAD1/5 activity and the transcriptional regulation of BMPs is still unknown.

### Staggered expression of Hox genes in *Nematostella* is directly controlled by BMP signaling

Bilateral body symmetry in *Nematostella* is manifested in the anatomy of the mesenteries subdividing the endoderm along the second body axis of the animal. In *Nematostella*, endodermal *hox* genes and *gbx* are expressed in staggered domains along the directive axis, and their expression boundaries exactly correspond to the positions of the emerging mesenteries (*Ryan et al., 2007*). This staggered endodermal *hox* and *gbx* expression is lost when BMP signaling is suppressed (*Genikhovich et al., 2015*). Recent loss-of-function analyses showed that RNAi of *hoxE, hoxD, hoxB*, or *gbx* led to the loss of the pairs of mesenteries corresponding to the expression boundaries of the knocked down genes, while the KD of the putative co-factor of all Hox proteins, *Pbx*, resulted in the loss of all mesenteries (*He et al., 2018*) phenocopying the loss of BMP signaling (*Genikhovich et al., 2015*; *Leclère and Rentzsch, 2014*; *Saina et al., 2009*). Our ChIP-seq analysis showed that the endodermal BMP-dependent staggered expression of *hox* genes and *gbx* is directly controlled by nuclear pSMAD1/5. In the future, it will be of great interest to test whether the loss of the mesenteries upon *BMP2/4, BMP5-8*, or *Chd* KD is only correlated with or actually caused by the resulting simultaneous suppression of all the staggered endodermal *hox* genes and *gbx* in the absence of BMP signaling (*Genikhovich et al., 2015*; *He et al., 2018*).

The question of whether Hox-dependent axial patterning was a feature of the cnidarian–bilaterian ancestor or evolved independently in Cnidaria and Bilateria remains debated. Various authors used different *hox* genes to homologize different cnidarian body axes to different bilaterian ones (*Arendt*

*et al., 2016*; *DuBuc et al., 2018*; *Finnerty et al., 2004*). The unclear orthology of cnidarian *hox* genes (*Chourrout et al., 2006*), as well as the staggered expression along a cnidarian body axis patterned by BMP signaling rather than along the one regulated by Wnt and FGF, as in Bilateria, allowed us to suggest that the involvement of staggered *hox* genes in axial patterning in anthozoans and bilaterians is probably convergent (*Genikhovich et al., 2015*; *Genikhovich and Technau, 2017*). This, however, does not exclude the possibility that regulation of some specific *hox* genes by BMP signaling was indeed conserved since before the cnidarian–bilaterian split. We find several *hox* genes as conserved direct BMP targets in *Xenopus* and *Drosophila*, and BMP-dependent regulation of *hox* expression has been reported in these models. For instance, in *Xenopus,* during the specification of the hindgut, direct pSMAD1/β-catenin interactions control the expression of *hoxa11, hoxb4,* and *hoxd1* (*Stevens et al., 2017*), and in *Drosophila*, a BMP-Hox gene regulatory network mutually regulating *decapentaplegic*, *labial,* and *deformed* was shown to be involved in the head morphogenesis (*Stultz et al., 2012*). Another indication in favor of the ancient origin of the BMP control over Hox-dependent processes is our finding that the gene encoding the TALE class homeodomain TF Meis is among the ChIP targets conserved between *Nematostella*, *Xenopus,* and *Drosophila*. Meis, together with another TALE class TF Pbx, forms a trimeric complex essential for the Hox and Parahox function in the bilaterian A-P axis patterning (*Merabet and Galliot, 2015*). Hox proteins were also shown to directly interact with Meis/Pbx in *Nematostella* (*Hudry et al., 2014*). This interaction appears to be crucial for the function of *Nematostella* Hox proteins since *Pbx* KD resulted in the loss of all eight mesenteries (*He et al., 2018*), phenocopying the 'no *hox*, no *gbx*' state of the endoderm with suppressed BMP signaling.

## Roles of BMP signaling during neurogenesis

In protostome and deuterostome Bilateria with centralized nervous systems, one of the key roles of BMP signaling is repression of neuroectoderm formation. In contrast, cnidarians possess diffuse nervous systems with certain local neural accumulations, which, however, cannot be considered ganglia or brains (*Arendt et al., 2016*; *Kelava et al., 2015*; *Martín-Durán and Hejnol, 2021*). There is no indication that the onset of nervous system development in *Nematostella* is affected by BMP signaling. The expression of neuronal terminal differentiation genes starts already at the blastula stage (*Richards and Rentzsch, 2014*), which is before the onset of detectable BMP signaling. During subsequent development, neurons continue to form in both germ layers in a radially symmetric manner (*Nakanishi et al., 2012*), regulated by Wnt, MAPK, and Notch signaling (*Layden et al., 2016*; *Layden and Martindale, 2014*; *Richards and Rentzsch, 2014*; *Richards and Rentzsch, 2015*; *Watanabe et al., 2014*). The only known exception to this is the population of GLWamide+ neurons. They arise on the *bmp2/4*-expressing side of the directive axis (i.e., in the area of minimal BMP signaling) under control of the atonal-related protein Arp6 (neuroD), which is the only *Nematostella* Arp gene with a bilaterally symmetric expression (*Watanabe et al., 2014*). Our ChIP and RNA-seq data showed that *Arp6* is directly suppressed by BMP (*Supplementary file 1*). *Arp6*, however, is not the only *Nematostella* BMP target gene whose bilaterian orthologs are implicated in the regulation of neuronal development. We also find *isl, irx, lmx, ashB, hmx3, atoh7,* and *soxC* – a *Sox4/Sox11* ortholog (the latter is also a BMP target in *Xenopus*) (*Bergsland et al., 2006*; *Doucet-Beaupré et al., 2015*; *Liang et al., 2011*; *Miesfeld et al., 2020*; *Rodríguez-Seguel et al., 2009*; *Stevens et al., 2017*; *Tomita et al., 2000*; *Wang et al., 2004*). Similarly, among the ChIP targets we find orthologs of the 'canonical' bilaterian axon guidance molecules such as *rgm*, *ephrin B*, and *netrin* (*Sun et al., 2011*; *Niederkofler et al., 2004*; *Williams et al., 2003*); however, it remains unclear whether these molecules, all of which have bilaterally symmetric expression in *Nematostella* (see *Figure 2* and *Matus et al., 2006a*), are involved in the regulation of neural development or have a different function. In Bilateria, these proteins are expressed in various non-neural contexts. For example, RGM was shown to act as a potentiator of BMP signaling in *Nematostella* and in Bilateria (*Leclère and Rentzsch, 2014*; *Mueller, 2015*), and Netrin was demonstrated to inhibit BMP signaling (*Abdullah et al., 2021*).

## *Nematostella* ZSWIM4-6 is a dampener of the pSMAD1/5 gradient and a potential conveyor of the BMP signaling-mediated gene repression

One of the most enriched pSMAD1/5 ChIP-targets in *Nematostella* was *zswim4-6*. The transcription of *Nematostella zswim4-6* exactly followed the dynamics of BMP signaling. Curiously, *zswim4-6*

expression appeared to be strongly affected only by the KD of the 'core BMP,' BMP2/4, while the effect of the KD of the 'modulator BMP,' GDF5-like on *zswim4-6* expression was minimal. We tested whether this apparent ligand-specific activation was due to BMP2/4 and GDF5-like using different type I BMP receptors (Alk2 vs. Alk3/6 or vice versa), but this hypothesis did not find experimental support. On the contrary, both Alk2 and Alk3/6 had to be knocked down in order to abolish *zswim4-6* transcription, suggesting that in our developmental context, Alk2 and Alk3/6 may need to heterodimerize to transduce BMP signal, similar to the situation reported in zebrafish (*Tajer et al., 2021*). Further analysis of the regulation of *zswim4-6* expression by BMP2/4 and GDF5-like suggested that *zswim4-6* was a highly sensitive BMP target, whose transcription could be initiated even by the residual BMP signal remaining after GDF5-like KD.

Members of the zinc-finger family with SWIM domain were found in Archaea, prokaryotes, and eukaryotes, and they were suggested to be capable of interacting with other proteins or DNA (*Makarova et al., 2002*). *Nematostella* ZSWIM4-6 is a nuclear protein, whose paralogs, ZSWIM4, ZSWIM5, and ZSWIM6, are conserved in Bilateria. To date, several studies have linked ZSWIM4/5/6 with vertebrate neurogenesis and forebrain development. For instance, in the mouse, the paralogs *zswim4-6* were found to be expressed in the forebrain with distinct spatiotemporal patterns (*Chang et al., 2020*; *Chang et al., 2021*). *Zswim6* knockout mice display abnormal striatal neuron morphology and motor function (*Tischfield et al., 2017*). Additionally, *zswim4* mutations frequently occur in patients with acute myelogenous leukemia (*Walter et al., 2012*), while *zswim6* mutations in mammals associate with acromelic frontonasal dysostosis (a rare disease characterized by craniofacial, brain and limb malformations), and schizophrenia (*Smith et al., 2014*; *Tischfield et al., 2017*). Interestingly, *zswim4/5* in the frog embryo and *zswim4/6* in transit-amplifying mouse hair follicle cells are also direct targets of BMP signaling (*Genander et al., 2014*; *Stevens et al., 2017*), and zebrafish *zswim5* is downregulated in gastrulating *bmp7*⁻/⁻ embryos of *D. rerio* (*Greenfeld et al., 2021*). This suggests that *zswim4-6* is a conserved downstream target of BMP signaling in several species and may contribute to BMP signaling-dependent patterning events in various contexts. A recent study on frog embryos and mammalian cells provided first glimpses into the function of Zswim4 highlighting its role in attenuating BMP signaling by directly interacting with and promoting ubiquitination and degradation of the nuclear SMAD1 (*Wang et al., 2022*). Our analyses of the effect of the up- or downregulation of the *Nematostella zswim4-6* and of the overexpression of the zebrafish *zswim5* on the shape of the pSMAD1/5 gradient in these two models appear to be in line with the vertebrate *zswim4* data (*Wang et al., 2022*): ZSWIM4-6 may indeed have a conserved function as a destabilizer of nuclear SMAD1/5. However, beyond the effect on the pSMAD1/5 gradient, analysis of the transcriptional response of the BMP signaling targets to *zswim4-6* KD in *Nematostella* revealed a more complex picture. We showed that the genes known to be negatively regulated by BMP signaling were de-repressed and expanded their expression toward the 'strong BMP signaling side' of the directive axis in spite of the increased levels of nuclear pSMAD1/5, while the genes positively regulated by BMP signaling were only weakly affected. Together with the results of our ChIP analysis of EGFP-tagged *Nematostella* ZSWIM4-6 showing that ZSWIM4-6 can bind to the pSMAD1/5 binding site in the upstream regulatory region of one pSMAD1/5-repressed gene, *chordin*, but not to the pSMAD1/5 binding site in the upstream regulatory region of one pSMAD1/5-activated gene, *gremlin*, this suggests that ZSWIM4-6 might act as a co-repressor of pSMAD1/5 targets. Currently, this interpretation is based on the expression changes of the BMP-dependent genes as well as on a very limited ChIP analysis. In the future, the generation of a transgenic line with a tag knocked-in into the endogenous *zswim4-6* locus followed by ChIP-seq will challenge the role of ZSWIM4-6 as a potential conveyor of the BMP-mediated transcriptional repression at the genome-wide scale.

## Conclusion

In this article, we demonstrate that in the sea anemone *N. vectensis*, BMP signaling directly controls the expression of a number of previously characterized and many novel regulators of their bilaterally symmetric body plan. We show that *gbx* and all the *hox* genes, previously shown to control the regionalization of the secondary body axis of *Nematostella,* are under direct BMP regulation. In our dataset, we identify BMP target genes conserved between Cnidaria and Bilateria, among which we found a novel modulator of BMP signaling, the dampener of the pSMAD1/5 gradient and putative transcriptional co-repressor *zswim4-6*, whose expression is tightly controlled by BMP signaling. While

the extracellular signaling network setting up the pSMAD1/5 gradient along the directive axis in *Nematostella* has been addressed before by us and others, ZSWIM4-6 is the first example of a nuclear protein modulating the pSMAD1/5 gradient in a non-bilaterian organism. Our article provides a valuable resource for future studies of BMP signaling in Cnidaria and Bilateria.

## Materials and methods

### Animal husbandry and microinjection

Adult *N. vectensis* polyps were kept in the dark at 18°C in *Nematostella* medium (16‰ artificial seawater, NM) and induced for spawning in a 25°C incubator with 10 hr of constant illumination. Egg packages were fertilized for 30 min, de-jellied in 3% L-cysteine/NM, and washed six times with NM. Microinjection was carried out under a Nikon TS100F Microscope using an Eppendorf Femtojet and Narishige micromanipulators as described in *Renfer and Technau, 2017*.

Adult TE zebrafish were kept in accordance with the guidelines of the EU directive 2010/63/EU and the German Animal Welfare Act as approved by the local authorities represented by the Regierungspräsidium Tübingen and the Regierungspräsidium Freiburg (Baden-Württemberg, Germany). Zebrafish embryos were maintained at 28°C in embryo medium (250 mg/l Instant Ocean salt in reverse osmosis water adjusted to pH 7 with NaHCO$_3$). Microinjections were carried out using PV820 Pneumatic PicoPumps (World Precision Instruments), M-152 micromanipulators (Narishige), and 1B100f-4 capillaries (World Precision Instruments) shaped with a P-1000 micropipette puller (Sutter Instrument Company).

### Transgenesis, gene overexpression, and knockdown

The full-length sequence of *Nematostella zswim4-6* was isolated by RACE-PCR. Native *zswim4-6* (*EF1α::zswim4-6-egfp*) and a truncated version lacking the first 24 bases coding for the NLS (*EF1α::zswim4-6Δnls-egfp*) were cloned into *AscI* and *SbfI* sites in the *Nematostella* transgenesis vector, downstream of the ubiquitously active *EF1α* promoter, and *I-SceI* meganuclease-mediated transgenesis was performed as specified (*Renfer and Technau, 2017*). For mRNA synthesis, the *EF1α* promoter was removed by digestion with *PacI* and *AscI*, the ends were blunted, and the plasmid re-ligated, which placed *zswim4-6* directly downstream of the SP6 promoter. Capped mRNA was synthesized using an SP6 mMessage mMachine Transcription Kit (Life Technologies) and purified with the Invitrogen MEGAclear Transcription Clean-Up Kit (Ambion). For overexpression, 150 ng/μl mRNA of *ZSWIM4-6-EGFP* or *EGFP* control were injected. *zswim4-6* knockout animals were generated using the IDT CRISPR/Cas9 (Alt-R CRISPR-Cas9 crRNA:tracrRNA) gRNA system. Genotyping of F0 mosaic mutants after in situ hybridization was performed as described in *Lebedeva et al., 2021*. Target sequence for gRNAs and genotyping primers is listed in *Supplementary file 3A*. Dominant-negative BMP receptors were generated by replacing the C-terminal serine-threonine kinase domains of Alk2, Alk3/6, and BMPRII with the *EGFP* coding sequence. mRNAs of dominant-negative receptors and *EGFP* control were injected at 60 ng/μl (*Supplementary file 3B*).

KD of *bmp2/4*, *chordin*, *gremlin,* and *gdf5-l* in *Nematostella* were performed using previously published morpholino oligonucleotides (*Supplementary file 3C*) as described (*Genikhovich et al., 2015*; *Saina et al., 2009*). The activity of the new translation-blocking ZSWIM4/6MO (5' CCGTCCAT AGCTTGTACTGATCGAC) was tested by co-injecting morpholino with mRNA containing the *mCherry* coding sequence preceded by either wild-type or 5-mismatch ZSWIM4/6MO recognition sequence in frame with *mCherry* (*Figure 6—figure supplement 1*, *Supplementary file 3D*). Approximately 4 pl (*Renfer and Technau, 2017*) of 500 μM solutions of GrmMO and GDF5-lMO, and 300 μM solutions of BMP2/4MO, ChdMO, and ZSWIM4/6MO were injected, unless indicated differently.

shRNA-mediated BMP receptor KD was carried out as described (*Karabulut et al., 2019*) using injection or electroporation of 800 ng/μl shRNA for *alk2*, *alk3/6*, and *bmprII* (*Supplementary file 3E*). shRNA against *mOrange* was used as control in all KD. To determine KD specificity, two non-overlapping shRNAs were tested for each gene and efficiencies were estimated by in situ hybridization and qPCR (*Supplementary file 3F*). qPCR normalization was performed using primers against GAPDH.

The zebrafish *zswim5* cDNA was cloned into the pCS2+ vector as follows. RNA was obtained from 75% epiboly-stage embryos using TRIzol (Invitrogen) and reverse-transcribed into cDNA using

SuperScript III First-Strand Synthesis SuperMix (Invitrogen). *zswim5* was then amplified from cDNA using the primers ATGGCGGAGGGACGTGGA and TTAACCGAAACGTTCCCGTACCA followed by the addition of *ClaI* and *EcoRI* restriction sites using the primers TTCTTTTTGCAGGATCCCATCGAT GCCACCATGGCGGAGGGACGTGGA and TAGAGGCTCGAGAGGCCTTGAATTCTTAACCGAAAC GTTCCCGTACCA. The resulting amplicon was cloned into pCS2+ using *ClaI* and *EcoRI* (NEB) digestion and Gibson assembly (NEB). To generate mRNA, the pCS2-zswim5 plasmid was linearized with *NotI*-HF (NEB) and purified using a Wizard SV Gel and PCR Clean-up System (Promega). mRNA was generated from linearized plasmid using an SP6 mMessage mMachine Transcription Kit (Thermo Fisher Scientific) and column-purified using an RNeasy Mini Kit (QIAGEN). mRNA concentration was quantified using an Implen NanoPhotometer NP80. For phenotype assessment, TE wild-type zebrafish embryos were injected through the chorion at the one-cell stage with 20, 40, 80, or 210 pg *zzswim5* mRNA. Unfertilized and damaged embryos were removed approximately 1.3 hr post-injection, and embryos were incubated at 28°C. At 1 d post-fertilization, embryos were scored based on gross morphology visible through the chorion. Representative embryos were immobilized with MESAB, dechorionated manually and imaged in 2% methylcellulose on an Axio Zoom V16 microscope (ZEISS).

All *Nematostella* KD and overexpression experiments were reproduced at least twice on biological replicates. For quantification of the phenotypes, embryos from the same biological replicate were used. *D. rerio zzswim5* overexpression experiments were performed on biological duplicates. For quantification, embryos from the same biological replicate were used. In case of both *Nematostella* and *Danio*, unfertilized eggs or damaged embryos were excluded from analysis of the phenotypes.

## Chromatin immunoprecipitation and RNA-seq

Processing of *Nematostella* embryos of two developmental stages, late gastrula and late planula, for ChIP was performed on biological triplicates as described previously (*Kreslavsky et al., 2017*; *Schwaiger et al., 2014*). Input DNA was taken aside after chromatin shearing and pre-blocking with Protein-G sepharose beads. Immunoprecipitation was performed overnight at 4°C using 300 μg of chromatin per reaction and polyclonal rabbit anti-Phospho-Smad1 (Ser463/465)/Smad5 (Ser463/465)/ Smad8 (Ser426/428) (Cell Signaling, #9511; 1:1000). Pulldown of the immunoprecipitated chromatin using protein-G sepharose beads and elution were performed as described (*Schwaiger et al., 2014*). Libraries for sequencing of the input and immunoprecipitated DNA were prepared using the NEBNext Ultra End Repair/dA-Tailing Module kit and NEBNextUltraLigation Module with subsequent PCR amplification using KAPA Real Time Library Amplification Kit. 50 bp PE Illumina HiSeq2500 sequencing was performed by the Vienna BioCenter Core Facilities.

Raw reads were trimmed using trimmomatic v0.32 (*Bolger et al., 2014*) using the paired end mode and the ILLUMINACLIP option, specifying TruSeq3-PE as the target adaptors, seed mismatches at 2, a palindrome clip threshold of 30, and a simple clip threshold of 10. Additionally, the leading and trailing quality threshold were 5, and reads under a length of 36 were filtered out. Trimmed reads were aligned to the *N. vectensis* genome (*Putnam et al., 2007*) using the 'mem' algorithm and default settings. Alignments were then prepared for conversion to bedpe format using samtools v1.3.1 (*Li et al., 2009*) name sort and fixmate utilities. Bedtools (*Quinlan and Hall, 2010*) was used to convert to bedpe. Peak calling was performed using a modified version of peakzilla (*Bardet et al., 2011*) ('peakzilla_qnorm_patched.py') which used quantile normalization for peak height. Peaks were merged across lanes using a custom script ('join_peaks.py') using a maximum distance of 100 bp. Joined peaks were then annotated using a custom script ('associate_genes.py'). Gene model assignment was then manually curated to discard wrong gene models. A filter was applied ('final_filter.py') which required the peaks to have an enrichment score of at least 10 and have an overall score of at least 80. Motif analysis was done using the MEME-ChIP pipeline (*Machanick and Bailey, 2011*) with the anr model, a window range of 5–30, number of motifs 10, DREME e-value threshold of 0.05, centrimo score and evalue thresholds of 5 and 10, respectively, against the JASPAR 2018 core motif database. Custom scripts mentioned above can be found in the following GitHub repository: https://github.com/nijibabulu/chip_tools (copy archived at *Zimmermann, 2024*) The overlap with the published *Nematostella* enhancers (*Schwaiger et al., 2014*) was identified using *bedtools intersect* function in bedtools v2.30.0 (*Quinlan and Hall, 2010*). The functional annotation of the anti-pSMAD1/5 ChIP target genes and of the 100 random NVE models was performed manually according to the NCBI BLASTX result (*Altschul et al., 1990*). One hundred random NVE models were

selected by randomly sorting all NVE models using the *sort -R* command in bash and taking the top 100 entries.

For RNA-seq, total RNA was extracted from 2 dpf embryos with TRIZOL (Invitrogen) according to the manufacturer's protocol. For each biological replicate and each experimental condition, >500 embryos were used. The number of the sequenced biological replicates is shown in *Figure 1—figure supplement 1*. Poly-A-enriched mRNA library preparation (Lexogen) and 50 bp SE Illumina HiSeq2500 sequencing was performed by the Vienna BioCenter Core Facilities. The reads were aligned with STAR (*Dobin et al., 2013*) to the *N. vectensis* genome (*Putnam et al., 2007*) using the ENCODE standard options, with the exception that `--alignIntronMax` was set to 100 kb. Hits to the NVE gene models v2.0 (https://doi.org/10.6084/m9.figshare.807696.v2) were quantified with featureCounts (*Liao et al., 2014*), and differential expression analysis was performed with DeSeq2 (*Love et al., 2014*). Expression changes in genes with the adjusted p-value<0.05 were considered significant. No additional expression fold change cutoff was imposed. Putative TFs and secreted molecules (SignalP) were identified using INTERPROSCAN (*Madeira et al., 2019*).

## Low-input ChIP

EGFP-IP was performed on *EF1α::zswim4-6-egfp* F0 transgenic *Nematostella* 30 hpf embryos. The experiment was performed on biological quadruplicates with 4000–6000 embryos per each replicate. Embryos were washed 2× with NM, 2× with ice-cold 1× PBS, and collected in a 2 ml tube. They were fixed in one part of 50 mM HEPES pH 8, 1 mM EDTA pH 8, 0.5 mM EGTA pH 8, 100 mM NaCl and three parts of 2% methanol-free formaldehyde (Sigma) in 1× PBS for 15 min at room temperature (RT) on an overhead rotator. The fixative was removed and exchanged with 125 mM glycine in PBS for 10 min and the embryos were washed 3× with PBS. For long-term storage, embryos were equilibrated in HEG buffer (50 mM HEPES pH 7.5, 1 mM EDTA, 20% glycerol), as much liquid as possible was removed, and embryos were snap-frozen in liquid nitrogen and stored at –80°. Before sonication, embryos were thawed on ice and resuspended in 1 ml E1 buffer (50 mM HEPES pH 7.5, 140 mM NaCl, 1 mM EDTA, 10% glycerol, 0.5% NP40, 0.25% Triton X-100, 1 mM PMSF, 1x cOmplete Protease Inhibitor [Roche], 1 mM DTT) and homogenized in a 1 ml dounce homogenizer with a tight pestle for 5 min directly on ice. The homogenate was centrifuged twice for 10 min at 4°C 1500 × *g*, the supernatant was discarde after the second centrifugation, and both pellets were resuspended in 130 µl lysis buffer (50 mM HEPES pH 7.5, 500 mM NaCl, 1 mM EDTA, 0.1% SDS, 0.5% N-laurylsarcosine sodium, 0.3% Triton X-100, 0.1% sodium deoxycholate, 1 mM PMSF, and 1x cOmplete Protease Inhibitor) and incubated in lysis buffer for 30 min. The samples were sonicated four times for 20 s in 130 µl glass tubes in a Covaris S220 with the following settings: peak power = 175, duty factor = 20, cycles/bursts = 200. with 9 s pausing steps (peak power = 2.5, duty factor = 0.1, cycles/bursts = 5) between each 20 s sonication round.

After sonication, the sample was transferred to a low-bind Eppendorf tube and centrifuged at 4°C for 10 min at 16,000 × *g*. For the input fraction, 15 µl of the sample were taken aside and stored at –20°C. For the IP, the rest of the sample was filled up to 1 ml with dilution buffer (10 mM Tris/HCl pH 7.5; 150 mM NaCl; 0.5 mM EDTA). GFP-Trap MA beads (Chromotek) were prepared according to the manufacturer's instructions and incubated with the sample at 4°C overnight under rotation. The samples were washed 1× with dilution buffer, 1× with high salt dilution buffer (500 mM NaCl), again with dilution buffer and eluted twice. Each elution was performed by adding 50 µl 0.2 M glycine pH 2.5, pipetting for 5 minutes. Supernatant was transferred to a fresh tube and neutralized by adding 10 µl 1 M Tris pH 8. The IP fractions were filled up to 200 µl with MilliQ water, and the input fractions were filled up to 200 µl with dilution buffer and incubated with 4 µl of 10 mg/ml RNase A for 30 min at 37°C. Then 10 mM EDTA pH 8.0, 180 mM NaCl, and 100 µg/ml Proteinase K were added and incubated overnight at 65°C. The samples were purified using a QIAGEN PCR Purification Kit. The IP fraction was eluted in 40 µl elution buffer, and the input fraction was eluted in 100 µl. The regulatory regions identified by pSMAD1/5 ChIP-Seq of *chordin* and *gremlin* were tested for enrichment by qPCR. Primers against the intergenic region A (*IntA*, scaffold_96:609598–609697) were used for normalization. Significance of the differences in the enrichments was assessed using the two-tailed *t*-test; however, the normality of the distribution of the data was assumed and not statistically confirmed due to the low number of datum points (four per each experimental condition).

## Orthology analysis

The *Nematostella* proteome was downloaded from a public repository (https://doi.org/10.6084/m9.figshare.807696.v2). The proteome from *D. melanogaster* version BDGP6, corresponding to that used in *Deignan et al., 2016*, was downloaded from FlyBase (*Larkin et al., 2021*). The *X. laevis* proteome version 9.1, corresponding to that used in *Stevens et al., 2017*, was downloaded from the JGI genome browser (*Nordberg et al., 2014*). Orthology links between all proteomes were inferred using NCBI blastp (*Altschul et al., 1990*), requiring an e-value of at most 1e-5. Reciprocal best blast hits were determined using the bit score as the metric. For *X. laevis*, the genome suffix (.S or.L) was ignored, and only genes with BLAST hits in all species were considered (*Figure 3*). The resulting hits were filtered for genes found as pSMAD1/5 targets in this study and in *Drosophila* and *Xenopus* analyses (*Deignan et al., 2016*; *Stevens et al., 2017*).

## In situ hybridization

Fragments for preparing in situ probes were amplified by PCR on cDNA (*Supplementary file 3G*) and cloned. In situ hybridization was carried out as described previously (*Lebedeva et al., 2021*), with minor changes. *Nematostella* embryos were fixed for 1 hr in 4% PFA/PTW (1× PBS, 0.1% Tween 20) at RT. Permeabilization was performed for 20 min in 10 µg/ml Proteinase K/PTW. 4d planulae were incubated for 40 min in 1 U/µl RNAseT1 in 2× SSC at 37°C after the initial 2× SSC wash and before the 0.075× SSC washes.

## Phylogenetic analysis

Protein sequences of ZSWIM4-6 and related proteins (*Supplementary file 2*) were aligned with MUSCLE using MegaX (*Kumar et al., 2018*) and trimmed with TrimAl 1.3 using an Automated 1 setting (*Capella-Gutiérrez et al., 2009*). A maximum likelihood tree (JTTDCMut+F+G4, bootstrap 100) was calculated using IQTREE (*Trifinopoulos et al., 2016*).

## Antibody staining and pSMAD1/5 quantification

Immunostainings in *Nematostella* were performed according to *Leclère and Rentzsch, 2014* with minor changes. Embryos were fixed for 2 min in ice-cold 0.25% glutaraldehyde/3.7% formaldehyde/PTX (1× PBS, 0.3% Triton-X), then transferred to ice-cold 3.7% FA/PTX and fixed at 4°C for 1 hr. The samples were washed 5× with PTX, incubated in ice-cold methanol for 8 min, and washed 3× with PTX. After 2 hr in the blocking solution containing 5% heat-inactivated sheep serum/95% PTX-BSA (PTX-BSA = 1% BSA/PTX), the samples were incubated overnight with the primary antibody (rabbit anti-pSMAD1/5/9 [Cell Signaling, #13820; 1:200] or rat αNvHoxE antibody [1:500]) in blocking solution. Embryos were then washed 5× with PTX, blocked for 2 hr in blocking solution, and incubated overnight with secondary antibody (goat α-rabbit IgG-Alexa568, goat α-rabbit IgG-Alexa633 or goat α-rat IgG-DyLight488) and DAPI (1:1000) in blocking solution. After the secondary antibody staining, the samples were washed 5× with PTX, mounted in Vectashield (VectorLabs), and imaged under Leica SP5X or Leica SP8 confocal microscopes. pSMAD1/5 staining intensities were quantified as described in *Genikhovich et al., 2015*. Morpholino and mRNA and *egfp* injected embryos were stained with anti-pSMAD1/5 and DAPI and 16-bit images of single confocal sections of oral views were collected. The intensities of the pSMAD1/5 staining of the nuclei were measured in a 180° arc of from 0 to π using Fiji (*Schindelin et al., 2012*). Measurements in the endodermal and the ectodermal body wall were performed separately. The nuclei were selected in the DAPI channel and measured in the pSMAD1/5 channel, starting in the middle of the domain where the pSMAD1/5 signal is strongest and moving toward the position on the opposite side of the embryo. Every nucleus in each embryo was assigned a coordinate along the 0 to π arc by dividing the sequential number of each nucleus by the total number of analyzed nuclei in that particular embryo. Anti-pSMAD1/5 staining intensities were then depicted as a function of the relative position of the nucleus.

TE wild-type zebrafish embryos were dechorionated using Pronase and injected at the one-cell stage with 20–210 pg *zswim5* mRNA (80 pg mRNA was injected for the quantification). Unfertilized and damaged embryos were removed ~1.5 hr post-injection, and embryos were incubated at 28°C. At 50% epiboly, embryos were fixed in cold 4% formaldehyde diluted in PBS. Uninjected embryos reached 50% epiboly 30–45 min before injected siblings. Fixed embryos were stored at 4°C overnight, washed with PBST, transferred to methanol (three PBST washes followed by three methanol washes),

and stored at –20°C for at least 2 hr. Embryos were then washed 4–5 times with PBST, blocked in FBS-PBST (10% FBS, 1% DMSO, 0.1% Tween-20 in PBS) for at least 1 hr, and then incubated in 1:100 anti-pSmad1/5/9 (Cell Signaling, #13820) in FBS-PBST at 4°C overnight. After 5–7 PBST washes over a period of 2–6 hr, embryos were incubated again in FBS-PBST for at least 1 hr and then transferred to 1:500 goat-anti-rabbit-Alexa488 (Life Technologies, A11008) and 1:1000 DRAQ7 (Invitrogen, D15105) overnight at 4°C. Embryos were washed 5–7 times with PBST over a period of 2–6 hr and stored at 4°C overnight. pSmad1/5/9 immunofluorescence and DRAQ7 nuclear signal were imaged on a Lightsheet Z.1 microscope (ZEISS). Embryos were mounted in 1% low melting point agarose and imaged in water using a ×20 objective. Embryos were oriented with the animal pole toward the imaging objective, and 50–90 z-slices with 7 µm between each slice were acquired. Quantification was performed on maximum intensity projections as described in *Rogers et al., 2020*. Briefly, the images were manually rotated in Fiji with the ventral side to the left, an ROI was manually drawn around each embryo to exclude non-embryo background, and the ventral-to-dorsal intensity profile was determined by calculating the average intensity in each column of pixels (pixel size: 0.457 µm × 0.457 µm). The profiles were normalized to % length to account for embryo-to-embryo variability in length and divided into bins of 0.5% embryo length. For background subtraction, intensities measured as described above in four randomly oriented controls not exposed to the primary antibody were subtracted from intensity profiles. Data from the first and last 5% of the body axis length were excluded from analysis because the averages in the dorsalmost and ventralmost regions are calculated on relatively few pixels and are, therefore, less reliable.

## Acknowledgements

This work was funded by the Austrian Science Foundation (FWF) grants P26962-B21 and P32705-B to GG and by the European Research Council (ERC) under the European Union's Horizon 2020 research and innovation program (grant agreement No 637840 [QUANTPATTERN] and 863952 [ACE-OF-SPACE]) to PM. We thank Michaela Schwaiger, Taras Kreslavsky, Hiromi Tagoh, and Patricio Ferrer Murguia for their help with the ChIP protocol, Matthias Richter and Christian Hofer for their assistance with in situ analyses, Emilio Gonzalez Morales for making the measurements for *Figure 6—figure supplement 3*, Catrin Weiler for the assistance in cloning zebrafish *zswim5*, David Mörsdorf for critically reading the manuscript and help with data visualization, and the Core Facility for Cell Imaging and Ultrastructure Research of the University of Vienna for access to the confocal microscope.

## Additional information

### Funding

| Funder | Grant reference number | Author |
|---|---|---|
| Austrian Science Fund | P26962-B21 | Grigory Genikhovich |
| Austrian Science Fund | P32705-B | Grigory Genikhovich |
| European Research Council | 637840 | Patrick Müller |
| European Research Council | 863952 | Patrick Müller |

The funders had no role in study design, data collection and interpretation, or the decision to submit the work for publication.

### Author contributions

Paul Knabl, Autumn P Pomreinke, Katherine W Rogers, Validation, Investigation, Visualization, Methodology, Writing – original draft, Writing – review and editing; Alexandra Schauer, Validation, Investigation, Visualization, Writing – original draft, Project administration, Writing – review and editing; Bob Zimmermann, Software, Formal analysis, Validation, Visualization, Writing – original draft, Writing – review and editing; Daniel Čapek, Validation, Investigation, Writing – review and editing; Patrick Müller, Conceptualization, Supervision, Funding acquisition, Writing – original draft, Project administration,

Writing – review and editing; Grigory Genikhovich, Conceptualization, Formal analysis, Supervision, Funding acquisition, Validation, Investigation, Visualization, Methodology, Writing – original draft, Project administration, Writing – review and editing

**Author ORCIDs**
Paul Knabl ⓘ http://orcid.org/0000-0001-8834-565X
Alexandra Schauer ⓘ https://orcid.org/0000-0001-7659-9142
Autumn P Pomreinke ⓘ http://orcid.org/0000-0001-6160-6426
Bob Zimmermann ⓘ http://orcid.org/0000-0003-2354-0408
Katherine W Rogers ⓘ https://orcid.org/0000-0001-5700-2662
Daniel Čapek ⓘ http://orcid.org/0000-0001-5199-9940
Patrick Müller ⓘ http://orcid.org/0000-0002-0702-6209
Grigory Genikhovich ⓘ http://orcid.org/0000-0003-4864-7770

**Decision letter and Author response**
Decision letter https://doi.org/10.7554/eLife.80803.sa1
Author response https://doi.org/10.7554/eLife.80803.sa2

---

## Additional files

**Supplementary files**
• Supplementary file 1. Anti-pSMAD1/5 ChIP analysis and RNA-seq analysis of BMP2/4 and GDF5-l morphants. Supplementary data 1 contains called peaks, curated gene list, and DE analysis of putative targets identified by anti-pSMAD1/5 ChIP. Sheet 1 (peak-calling-LG) and sheet 2 (peak-calling-4dP) contain number and positions of identified peaks at the late gastrula (LG) and in the 4 d planula (4dP) by anti-pSMAD1/5 ChIP, respectively. Sheet 3 (LG4dP-targets) depicts the curated list of putative pSMAD1/5 targets at LG and 4dP combined. Sheet 4 (BMP24MO2dvsStdMO2d-padj<0.05) and sheet 5 (GDF5lMO2dvsStdMO2d-$p_{adj}$<0.05) contain differentially expressed genes upon morpholino knockdown of BMP2/4 and GDF5-l with $p_{adj}$ = 0.05. Sheet 6 (ChIP-RNA-Seq-overlap_2x) displays pSMAD1/5 targets that are differentially expressed in BMP2/4 morphants and GDF5-l morphants (2×-fold, $p_{adj}$ = 0.05) and sheet 7 (enrichment-of-BRE-in-funct-cat) displays DE pSMAD1/5 targets containing the BRE motif.

• Supplementary file 2. Trimmed alignment of ZSWIM proteins.

• Supplementary file 3. RNA, DNA and Morpholino oligonucleotide sequences; shRNA target sites. (**A**) ZSWIM4-6 guide RNA sequences and sequencing primers. (**B**) Primers used for cloning dominant-negative BMP receptor fragments. (**C**) Morpholino sequences. (**D**) Primer sequences for MO specificity testing. (**E**) Short hairpin RNA targets. (**F**) qPCR primers. (**G**) Primer sequences for in situ probes.

• MDAR checklist

### Data availability
The data generated or analysed during this study are included in the manuscript and supporting files. Source data files have been provided for Figure 6D-F', Figure 6Q, Figure 5—figure supplement 1D, Figure 6—figure supplement 2G, Figure 6—figure supplement 3E and F. All the sequencing data are available at NCBI as an SRA project PRJNA820593.

The following dataset was generated:

| Author(s) | Year | Dataset title | Dataset URL | Database and Identifier |
|---|---|---|---|---|
| Schauer A, Genikhovich G | 2022 | BMP signaling targets in Nematostella | https://www.ncbi.nlm.nih.gov/bioproject/PRJNA820593 | NCBI BioProject, PRJNA820593 |

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
