## [Editor Report]

This important work presents a systematic survey of downstream target genes of the BMP pathway during body-axis establishment of the cnidarian *Nematostella vectensis*. Combining genomic approaches and genetic manipulations across species, the authors present convincing evidence that Zswim4-6 acts as a negative feedback regulator of BMP activity whose function is evolutionarily conserved. Thus, this work will be of interest to researchers in both developmental biology and evo-devo.

---

## [Decision Letter]

**Decision letter after peer review:**

Thank you for submitting your article "Analysis of SMAD1/5 target genes in a sea anemone reveals ZSWIM4-6 as a novel BMP signaling modulator" for consideration by *eLife*. Your article has been reviewed by 3 peer reviewers, and the evaluation has been overseen by a Reviewing Editor and Marianne Bronner as the Senior Editor. The following individuals involved in the review of your submission have agreed to reveal their identity: Chiara Sinigaglia (Reviewer #1); Shuonan He (Reviewer #3).

Essential revisions:

1) The authors should more clearly discuss the possible function of zswim4-6. There seem to 3 different proposed roles (all of which are interesting) between the two animals analyzed, but none of these is particularly well supported: (1) zswim4-6 modifies the BMP gradient (based on MO loss of function in Nv with pSmad staining lower in endoderm and higher in ectoderm), (2) zswim4-6 acts as a pSmad co-repressor (based on limited qChIP-PCR on two pSmad target genes in Nv), (3) zswim5 broadly dampens the BMP gradient (based on zebrafish gain of function showing loss of pSmad staining across the embryo and some phenotypes that may be related to BMP loss of function). Since zswim is a major focus of the manuscript, the authors should be clearer about what it is doing with respect to BMP signaling and BMP target gene regulation. They should also clearly discuss the limitations of the experiments and conclusions. Although in general not required by *eLife* policy, limited further experiments should help clarify the function of zswim in some of the cases, and more evidence for any one of the three possible roles would significantly improve the manuscript.

2) The conclusion that Zswim4-6 is a conserved BMP regulator is based on cross-species comparisons of BMP direct targets and the finding that Zswim4-6 is a BMP target both in Nematostella and *Xenopus*. However, the subsequent functional validation was done in zebrafish, whose Zswim4-6 homologs may or may not be directly regulated by BMP. As suggested by Reviewer#2, the authors should also look at published data from zebrafish during the comparison and check whether Zzswim5 is also subjected to BMP regulation.

3) The authors claim that Zzswim5 mRNA injection mimics BMP mutations. Two reviewers found that the evidence to support this was not compelling. The authors may consider a limited set of experiments such as carrying out in situ hybridization of conventional zebrafish DV marker genes such as Chd, Szl and Vent in Zzswim5 injected zebrafish larvae. Alternatively, the authors may choose to tune down the conclusion regarding the conserved function of Zswim4-6 in both the abstract as well as the discussion.

4) A major focus of the manuscript is to dissect the intimate interactions between BMP signal and Zswim4-6 during Nematostella directive axis establishment. However, the in situ patterns of Zswim4-6 are variable in control panels between figures 2, 4, 5, and 6, making it difficult to interpret the effects of different experimental manipulations. The authors should either explain the reason for this variability or provide more consistent Zswim4-6 in situ results to allow meaningful comparisons between experiments. The authors should also address the differential responses of Zswim4-6 in the ectoderm versus the endoderm, as suggested by Reviewer #2.

5) The additional comments made by the reviewers could help to improve the clarity of the manuscript and should be considered by the authors.

*Reviewer #1 (Recommendations for the authors):*

i) I recommend avoiding the term "Protostome" (pg 11, Figure 1A), and opting for Lophotrocozoa and Ecdysozoa instead.

ii) The 3 days old planula in Figure 1D shows asymmetric staining for pSMAD1/5 activity (perpendicular to the directive axis): is this biologically relevant?

iii) Figure 2: the staging of 2d old planulae appears to vary. Is this reflecting a developmental variability?

iv) The expression patterns for zswim4-6 at the 4-day planula stage appear slightly different between Figure 2 and Figure 4F. In particular, the pattern shown in Figure 2 could be interpreted as being still bilaterally expressed.

v) The ZSWIM *Xenopus* sequences were not included in the phylogeny, I would suggest presenting them in Figure 4A for completeness.

vi) In Figures 6A and B, please display the level in the polyp body column at which the section was taken.

vii) With reference to Figure 6O-O' the text reads: "not much change was observed in the gdf5-like expression", however from that image, it seems that the domain of expression is expanded. Could the Authors comment?

viii) Page 12: please add a reference to data after the statement "Our ChiP and RNASeq data showed that Arp6 is directly suppressed by BMP".

*Reviewer #2 (Recommendations for the authors):*

1. In the analysis of enrichment of different type of BMP target genes, how were the 100 random genes selected?

2. In the analysis of shared direct BMP targets between three species, I didn't follow the anchoring analysis that was able to reveal additional direct targets shared between all three species. Is this based on there being a single gene in Nematostella that was duplicated during evolution such that there are multiple homologues in other species (gata for instance)?

*Reviewer #3 (Recommendations for the authors):*

1. pSMAD1/5 ChIP-seq is a major highlight of the manuscript and serves as the foundation for subsequent analysis. In Figure 1 or the related supplemental figure, the authors should at least provide examples of genome browser views at certain target loci to demonstrate the specificity of the antibody used as well as the quality of their data. Showing representative pSMAD1/5 binding profiles at two developmental stages should also enable the audience to better visualize the dynamic changes of BMP activity at certain downstream targets and help illustrate the presence of BRE elements in relation to the pSMAD1/5 peaks.

2. Schwaiger and colleagues (2014) have systematically examined histone modifications, PolII, and p300 binding profiles at gastrula and planula stages in Nematostella. It would be informative to quantify the percentage of pSMAD1/5 peaks overlapping with previously annotated enhancer/open chromatin features.

3. The authors did not observe pSMAD1/5 peaks closely associated with BMP ligand-coding genes BMP2/4, BMP5-8, and GDF5-l, and concluded that these genes are not direct targets of BMP signal. This is surprising given previous observations using BMP MO-injected larvae in which GDF5-l expression is abolished and BMP2/4 and BMP5-8 expression increased (Genikhovich, et al., 2015). Do the authors observe similar expression changes in their RNA-seq dataset? If so, what could be the explanation? Is it possible that enhancer elements of these genes are located distally and as a result, the pSMAD1/5 peaks are assigned to other genes? Or maybe indirect gene regulation is in play here?

4. As stated in the public review section, I'm not sure if Figure 3 adds much to the overall story by forcefully comparing three datasets from different lineages at different developmental stages. As the authors stated themselves, across such large evolutionary distances, homology between genes such as Hox genes can be extremely challenging to discern. Even in cases where homology is evident, one cannot distinguish between ancestral regulatory interaction that was retained versus convergent evolution. I think a simple statement that many BMP targets identified in Nematostella have also been implicated to be regulated by BMP in different bilaterian lineages should be sufficient to convey the message. In addition, the left most and right most columns of transcription factor names in the Xla/Dme comparison are duplicated. Please correct if the authors intend to keep this figure.

5. The authors need to provide more consistent control images of Zswim4-6 in situ results in figure 5 and figure 6. In figure 5 B', H, and M, Zswim4-6 expression level and expression patterns (especially in the ectoderm) are clearly variable between control MO injection, EGFP mRNA injection, and sh_mOr injections. Is this due to stage differences when the larvae are collected or different in situ developing time? Since the goal here is to compare the effect of BMP manipulations on Zswim4-6 expression, more consistent control images are necessary. Otherwise, comparing Figure 5 panel H and I will lead to the conclusion that injecting dn_alk2 mRNA specifically activates Zswim4-6 expression in the ectoderm, opposite to what the authors are trying to claim.

6. Similar issues exist for Fig6 G to P, and the expression patterns of BMP2/4, Zswim4-6 and Gremlin are not comparable between control MO and EGFP injected larvae, which as the authors explained, is due to collection timing differences caused by rapid degradation of injected Zswim4-6 mRNA. To avoid confusion, maybe the authors can add stage labels (late gastrula versus mid-gastrula, or specific hours post fertilization) on top of these panels to guide the naïve audience.

---

## [Author Response]

Essential revisions:1) The authors should more clearly discuss the possible function of zswim4-6. There seem to 3 different proposed roles (all of which are interesting) between the two animals analyzed, but none of these is particularly well supported: (1) zswim4-6 modifies the BMP gradient (based on MO loss of function in Nv with pSmad staining lower in endoderm and higher in ectoderm), (2) zswim4-6 acts as a pSmad co-repressor (based on limited qChIP-PCR on two pSmad target genes in Nv), (3) zswim5 broadly dampens the BMP gradient (based on zebrafish gain of function showing loss of pSmad staining across the embryo and some phenotypes that may be related to BMP loss of function). Since zswim is a major focus of the manuscript, the authors should be clearer about what it is doing with respect to BMP signaling and BMP target gene regulation. They should also clearly discuss the limitations of the experiments and conclusions. Although in general not required by eLife policy, limited further experiments should help clarify the function of zswim in some of the cases, and more evidence for any one of the three possible roles would significantly improve the manuscript.

We agree that our current evidence supports at least two non-mutually exclusive functions of the *Nematostella* ZSWIM4-6: (i) as modulator of the pSMAD1/5 gradient, and (ii) as a potential transcriptional repressor of the genes negatively regulated by BMP signaling.

The role of the ZSWIM4-6 in dampening the pSMAD1/5 gradient in *Nematostella* is in line with our observations of the effect of the overexpression of *zswim5* in zebrafish (listed as option 3 in your comment above) as well as with the data from the recent preprint of Wang et al., 2022. There, using frog embryos and mammalian cells, they show that Zswim4 directly interacts with SMAD1 leading to its ubiquitination and degradation. To further address the possibility that ZSWIM4-6 may be involved in destabilizing pSMAD1/5 we co-injected ZSWIM4-6MO together with the GDF5-likeMO. GDF5-like knockdown drastically reduces pSMAD1/5 levels without abolishing it (Genikhovich et al. 2015), however, co-injection of ZSWIM4-6MO with the GDF5-likeMO partially rescues this effect, in line with the vertebrate data (Wang et al., 2022). The results of this experiment are shown in the new Figure 6 —figure supplement 3. While the precise molecular mechanism by which *Nematostella* ZSWIM4-6 exerts its function will require further analysis, its anti-pSMAD1/5 effect appears quite clear.

Testing the role of ZSWIM4-6 as a co-repressor of BMP-repressed genes proved to be more difficult than anticipated. An appropriate analysis of this would be at a genome-wide level by ChIP-seq identification of ZSWIM4-6-bound genes, however, this would require either raising anti-ZSWIM4-6 antibodies or knocking-in a tag into the *zswim4-6* locus, which is beyond the scope of this work. As a second approach, we tried to overactivate BMP signaling activity with concomitant ZSWIM4-6 knockdown to further test whether BMP-repressed targets become de-repressed in absence of

ZSWIM4-6. Unfortunately, we could not achieve a reliable, long-term upregulation of the BMP signaling activity by either injecting *Nematostella bmp* mRNAs or treating the embryos with human recombinant BMP2 and BMP5. In the discussion, we have stressed that the conclusion of ZSWIM4-6 being a co-repressor of pSmad5 targets is based on expression changes of the known BMP targets and on the direct ChIP analysis of two target genes.

2) The conclusion that Zswim4-6 is a conserved BMP regulator is based on cross-species comparisons of BMP direct targets and the finding that Zswim4-6 is a BMP target both in Nematostella and *Xenopus*. However, the subsequent functional validation was done in zebrafish, whose Zswim4-6 homologs may or may not be directly regulated by BMP. As suggested by Reviewer#2, the authors should also look at published data from zebrafish during the comparison and check whether Zzswim5 is also subjected to BMP regulation.

The comparison was done between *Nematostella*, *Xenopus* and *Drosophila* because pSMAD1/5 ChIP-seq data only exist for these models. The functional validation was performed in zebrafish because we were planning to generate zswim5 mutants, whose analysis is currently in progress. We looked at the RNA-Seq data from zebrafish as suggested by the Reviewer 2, and found that *zzswim5* was downregulated in gastrulating *bmp7* mutant zebrafish, however, this gene was not among the 57 genes, which were considered to be direct BMP targets because their expression was affected by *bmp7* mRNA injection into cycloheximide-treated *bmp7* mutants (Greenfeld et al., 2021).

3) The authors claim that Zzswim5 mRNA injection mimics BMP mutations. Two reviewers found that the evidence to support this was not compelling. The authors may consider a limited set of experiments such as carrying out in situ hybridization of conventional zebrafish DV marker genes such as Chd, Szl and Vent in Zzswim5 injected zebrafish larvae. Alternatively, the authors may choose to tune down the conclusion regarding the conserved function of Zswim4-6 in both the abstract as well as the discussion.

We performed in situ hybridization with the dorsal and ventral markers as suggested, but the results were not conclusive. We therefore tuned down all our zebrafish arguments limiting ourselves to noting that the effect of *zzswim5* overexpression reduces pSMAD1/5, as is also the case in *Nematostella* and, according to the work of Wang et al., 2022, in the frog.

4) A major focus of the manuscript is to dissect the intimate interactions between BMP signal and Zswim4-6 during Nematostella directive axis establishment. However, the in situ patterns of Zswim4-6 are variable in control panels between figures 2, 4, 5, and 6, making it difficult to interpret the effects of different experimental manipulations. The authors should either explain the reason for this variability or provide more consistent Zswim4-6 in situ results to allow meaningful comparisons between experiments. The authors should also address the differential responses of Zswim4-6 in the ectoderm versus the endoderm, as suggested by Reviewer #2.

The mentioned variability of in situ patterns is due to multiple changes and general optimization of the in situ protocol over the last years. To avoid confusion, we repeated the in situs for all figures mentioned above if they were performed with the old versions of the in situ protocol.

5) The additional comments made by the reviewers could help to improve the clarity of the manuscript and should be considered by the authors.

Please see responses to the comments of the Reviewers below.

Reviewer #1 (Recommendations for the authors):i) I recommend avoiding the term "Protostome" (pg 11, Figure 1A), and opting for Lophotrocozoa and Ecdysozoa instead.

While we agree that the term Protostomia makes little sense embryologically, we believe that it makes sense to use the term as phylogenetically it is a convenient monophyletic group encompassing Lophotrochozoa and Ecdysozoa under a single term.

ii) The 3 days old planula in Figure 1D shows asymmetric staining for pSMAD1/5 activity (perpendicular to the directive axis): is this biologically relevant?

At 3 days post fertilization, pSMAD1/5 staining is always quite variable in *Nematostella*. During this time *chordin* expression starts to dwindle and the gradient starts to disappear. We do not think that the asymmetry you mentioned is biologically relevant but rather represents a transition between the graded signaling we observe at 2 dpf and signaling in all 8 mesenteries we observe at 4 dpf.

iii) Figure 2: the staging of 2d old planulae appears to vary. Is this reflecting a developmental variability?

While there is always a certain variability in the developmental rates of the embryos, we do not believe that the differences are drastic and the embryos look quite comparable.

iv) The expression patterns for zswim4-6 at the 4-day planula stage appear slightly different between Figure 2 and Figure 4F. In particular, the pattern shown in Figure 2 could be interpreted as being still bilaterally expressed.

Thanks for noticing this. As the planula on Figure 2 was slightly overstained, we replaced the image with a better representative example.

v) The ZSWIM *Xenopus* sequences were not included in the phylogeny, I would suggest presenting them in Figure 4A for completeness.

We now added *Xenopus* sequences to the tree as well (Figure 4A). Thanks for the suggestion!

vi) In Figures 6A and B, please display the level in the polyp body column at which the section was taken.

Phalloidin and anti-HoxE stainings were performed not on polyps but on 4 dpf planulae, as indicated in the bottom right corner on the image. We have added a schematic to indicate the approximate plane at which the optical section was taken (Figure 6A,B).

vii) With reference to Figure 6O-O' the text reads: "not much change was observed in the gdf5-like expression", however from that image, it seems that the domain of expression is expanded. Could the Authors comment?

Thanks for noticing this. In controls, there is a clear strong GDF5-like side, while upon *zswim4-6* mRNA injection, the staining becomes weaker. We repeated the in situ, replaced the control image with a more representative one and corrected the text. The reduction of *gdf5-like* and *grm* reflect the reduction of the pSMAD1/5 levels.

viii) Page 12: please add a reference to data after the statement "Our ChiP and RNASeq data showed that Arp6 is directly suppressed by BMP".

We added a reference to the Supplementary table 1.

Reviewer #2 (Recommendations for the authors):1. In the analysis of enrichment of different type of BMP target genes, how were the 100 random genes selected?

We added the following explanation:

"100 random NVE models

(https://figshare.com/articles/Nematostella_vectensis_transcriptome_and_gene_models _v2_0/807696) were selected by randomly sorting the NVE models using the sort -R command in bash and taking the top 100 entries. The models were then manually annotated based on the NCBI BLASTX result".

2. In the analysis of shared direct BMP targets between three species, I didn't follow the anchoring analysis that was able to reveal additional direct targets shared between all three species. Is this based on there being a single gene in Nematostella that was duplicated during evolution such that there are multiple homologues in other species (gata for instance)?

Yes, exactly. We re-wrote this part of the Results section to make it clearer.

Reviewer #3 (Recommendations for the authors):1. pSMAD1/5 ChIP-seq is a major highlight of the manuscript and serves as the foundation for subsequent analysis. In Figure 1 or the related supplemental figure, the authors should at least provide examples of genome browser views at certain target loci to demonstrate the specificity of the antibody used as well as the quality of their data. Showing representative pSMAD1/5 binding profiles at two developmental stages should also enable the audience to better visualize the dynamic changes of BMP activity at certain downstream targets and help illustrate the presence of BRE elements in relation to the pSMAD1/5 peaks.

Thanks for the suggestion. We added a new Figure-1—figure supplement-1 to show examples.

2. Schwaiger and colleagues (2014) have systematically examined histone modifications, PolII, and p300 binding profiles at gastrula and planula stages in Nematostella. It would be informative to quantify the percentage of pSMAD1/5 peaks overlapping with previously annotated enhancer/open chromatin features.

Thanks for the suggestion. We added this information to the text and to the Supplementary Data 1 file.

3. The authors did not observe pSMAD1/5 peaks closely associated with BMP ligand-coding genes BMP2/4, BMP5-8, and GDF5-l, and concluded that these genes are not direct targets of BMP signal. This is surprising given previous observations using BMP MO-injected larvae in which GDF5-l expression is abolished and BMP2/4 and BMP5-8 expression increased (Genikhovich, et al., 2015). Do the authors observe similar expression changes in their RNA-seq dataset? If so, what could be the explanation? Is it possible that enhancer elements of these genes are located distally and as a result, the pSMAD1/5 peaks are assigned to other genes? Or maybe indirect gene regulation is in play here?

We agree that the lack of pSMAD1/5 binding may seem surprising given that expression of *Nematostella* BMP genes is affected by BMP knockdown, and in *Drosophila* and frog nearly all BMP genes are directly regulated by BMP signaling.

Expression changes in the RNA-Seq dataset mostly fit previous in situ hybridization results from us and others. For example, in the BMP2/4 KD dataset, *Chordin*, *rgm*, *bmp2/4*, *bmp5-8* are upregulated, and *Gremlin*, *Gbx*, *HoxE/Anthox1a*, *HoxD/Anthox8*, *HoxB/Anthox6a*, *Lbx* are downregulated. Of all the genes with the previously documented expression change upon the BMP2/4 KD, only *GDF5-like* did not show a statistically significant downregulation in the RNA-Seq, although it was abolished in the in situ hybridization (Genikhovich et al., 2015). Similarly, upon GDF5-like KD, *Gremlin*, *HoxE* and *HoxD* were downregulated, while changes in the *HoxB* and *Gbx* were not significant, as observed also in earlier in situ results (Genikhovich et al., 2015).

Certainly, it is possible that the expression of the BMP genes is controlled by distal pSMAD1/5-bound enhancers, but the necessary chromosome conformation capture experiments to test this hypothesis are out of scope of this paper.

We believe that indirect regulation is highly probable. First of all, we have generated a transgenic *bmp2/4::mCherry* reporter line (Figure 6D-D’ in Schwaiger et al., 2014, https://doi.org/10.1101/gr.162529.113), which shows expression on one side of the directive axis of the early embryo. This expression is driven by a 4.5 kb fragment of gDNA upstream of the translation start site in the *bmp2/4*, and our ChIP shows no significant pSMAD1/5 binding in this region. Second, once *chordin* stops to be expressed by 4d post fertilization, we observe BMP signaling in the mesenteries, i.e. in the same cells, which express *bmp2/4* and *bmp5-8*. Since our KD experiments clearly show that both these BMP genes are suppressed by BMP signaling in the 1d and 2d embryos, it is likely that in 1d and 2d embryos this repression is mediated not by pSMAD1/5 but by some other factor, which is positively regulated by pSMAD1/5 and which is not expressed in the mesenteries of the 4d planula.

4. As stated in the public review section, I'm not sure if Figure 3 adds much to the overall story by forcefully comparing three datasets from different lineages at different developmental stages. As the authors stated themselves, across such large evolutionary distances, homology between genes such as Hox genes can be extremely challenging to discern. Even in cases where homology is evident, one cannot distinguish between ancestral regulatory interaction that was retained versus convergent evolution. I think a simple statement that many BMP targets identified in Nematostella have also been implicated to be regulated by BMP in different bilaterian lineages should be sufficient to convey the message. In addition, the left most and right most columns of transcription factor names in the Xla/Dme comparison are duplicated. Please correct if the authors intend to keep this figure.

Thank you very much for noticing the error in the Figure 3! We have corrected it. As to your comment regarding the content of the figure, we agree that orthology statements for the individual cnidarian and bilaterian Hox genes do not have factual support. Therefore, we removed a Hox gene from the list of targets potentially conserved between all three species. Nevertheless, given the abundance of BMP targets among *Nematostella*, *Drosophila* and *Xenopus* Hox genes, there is a certain probability that one or more Hox genes of the last common cnidarian-bilaterian ancestor was under BMP control. The fact that the Hox co-factor Meis is also a shared target for all three species adds extra weight to this probability. For the other genes, orthology statements are less problematic, and we think it important to point out which genes are repeatedly recovered as direct BMP targets in these very distantly related species. We certainly cannot exclude the possibility that this may be a result of convergence rather than conservation, but we might see more support for either conservation or convergence hypothesis once BMP targets are identified in a wider variety of animals across Planulozoa.

5. The authors need to provide more consistent control images of Zswim4-6 in situ results in figure 5 and figure 6. In figure 5 B', H, and M, Zswim4-6 expression level and expression patterns (especially in the ectoderm) are clearly variable between control MO injection, EGFP mRNA injection, and sh_mOr injections. Is this due to stage differences when the larvae are collected or different in situ developing time? Since the goal here is to compare the effect of BMP manipulations on Zswim4-6 expression, more consistent control images are necessary. Otherwise, comparing Figure 5 panel H and I will lead to the conclusion that injecting dn_alk2 mRNA specifically activates Zswim4-6 expression in the ectoderm, opposite to what the authors are trying to claim.

Please see comments to the Editor. The differences are due to changes in our in situ protocol. We have repeated the experiments using the new in situ protocol and replaced the problematic images.

6. Similar issues exist for Fig6 G to P, and the expression patterns of BMP2/4, Zswim4-6 and Gremlin are not comparable between control MO and EGFP injected larvae, which as the authors explained, is due to collection timing differences caused by rapid degradation of injected Zswim4-6 mRNA. To avoid confusion, maybe the authors can add stage labels (late gastrula versus mid-gastrula, or specific hours post fertilization) on top of these panels to guide the naïve audience.

Please see the responses to the Essential revisions. The differences are due to changes in our in situ protocol. We have repeated the experiments using the new in situ protocol and replaced the problematic images.